# Embodied-DETR: End-to-End Temporal 3D Object Detection in Egocentric Views

Ziheng Ding [1]  Xiaze Zhang [1]  Yuejie Zhang [1]  Lifeng Chen [1]  Rui Feng [1 2]

## Abstract

Embodied 3D object detection is a fundamental perceptual capability for embodied agents, in which observations are partial, heavily occluded, and sequential, requiring modeling of temporal continuity. However, existing benchmarks and methods are primarily designed for fully reconstructed global scenes and fail to capture temporal observation context and instance evolution in first-person perception. We introduce *Embodied-Det*, a new benchmark for embodied 3D object detection that evaluates detection accuracy, temporal stability, and consistency under egocentric sequential views. Building on this benchmark, we propose *Embodied-DETR*, an end-to-end temporal detection framework that models scene-level context and instance-level consistency through two complementary temporal modules, **Scene-aware Feature Aggregation** and **Instance-aware Query Embedding**. Experiments on *Embodied-Det* show that existing methods suffer substantial performance degradation in egocentric temporal settings, while *Embodied-DETR* achieves superior accuracy and temporal consistency, demonstrating the effectiveness of temporal modeling for embodied 3D perception. Codes are available at https://github.com/UniPerceptor/UniPerceptor.

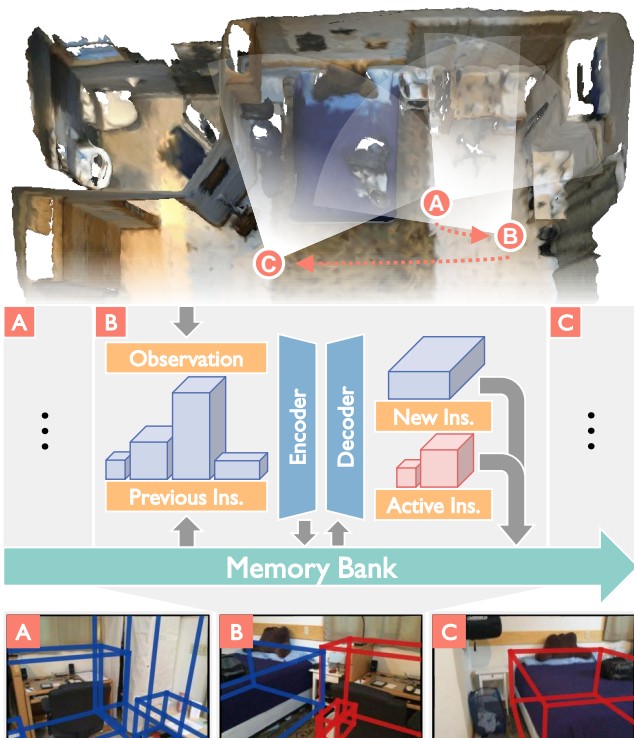

*Figure 1.* Example of embodied detection scenario. The robot traverses A, B, and C sequentially while observing from an egocentric view. Our method leverages previously detected instances to aid current detection. Blue boxes indicate predicted newborn instances, while red boxes indicate that the model detects seen instances using temporal information. Best viewed in color.

## 1. Introduction

3D object detection is a core task in computer vision, underpinning a wide range of robotics applications. It enables embodied intelligent agents to perceive and understand ob-

[1]College of Computer Science and Artificial Intelligence, Shanghai Key Laboratory of Intelligent Information Processing, Fudan University [2]College of Intelligent Robotics and Advanced Manufacturing, Fudan University. Correspondence to: Rui Feng <fengrui@fudan.edu.cn>.

*Proceedings of the $43^{rd}$ International Conference on Machine Learning*, Seoul, South Korea. PMLR 306, 2026. Copyright 2026 by the author(s).

jects in their surrounding environment. Significant research has focused on indoor 3D object detection, including pioneering voting-based strategies (Qi et al., 2019; Zhang et al., 2020), efficient architectures (Rukhovich et al., 2022; 2023), and 3D Detection Transformer (DETR (Carion et al., 2020)) framworks (Liu et al., 2021; Misra et al., 2021; Shen et al., 2024; Kolodiazhnyi et al., 2025). In terms of benchmarks, existing efforts have collected large-scale, diverse indoor scenes (Dai et al., 2017; Armeni et al., 2017; Chang et al., 2017), typically represented as fully reconstructed room layouts containing multiple objects.

However, a significant gap persists between these estab-

lished paradigms and the operational reality of embodied agents. First, conventional methods typically assume access to meticulously reconstructed scenes, enabling holistic perception from a global view and producing all detections in a single pass. In contrast, an embodied agent must rely on sequential, egocentric observations from its onboard sensors (*e.g.*, an RGB-D camera), which are inherently partial, occluded, and noise-corrupted. Second, while embodied interactions are continuous and exhibit strong spatiotemporal correlations, prevailing methods perform only single-frame detection, thereby discarding crucial temporal context. In reality, historical observations can substantially enhance future detection under sequential egocentric perception.

To bridge this gap, we first introduce *Embodied-Det*, a novel benchmark for embodied 3D object detection. Unlike standard indoor 3D detection that operates on fully reconstructed scenes, *Embodied-Det* targets the sequential, egocentric perception setting of embodied agents, where only partial, view-dependent observations are available at each timestep. It provides a multi-dimensional evaluation protocol that assesses detection accuracy as well as temporal consistency and stability, capturing the reliability of perception over time. *Embodied-Det* is constructed by reorganizing the widely-used ScanNet v2 dataset (Dai et al., 2017), leveraging its original frame sequences and reconstructed scenes to derive per-frame annotations for online temporal detection.

Building upon this benchmark, we propose *Embodied-DETR*, an end-to-end temporal detection transformer for embodied 3D perception. *Embodied-DETR* leverages multi-frame observations effectively through two core temporal modules: *Scene-aware Feature Aggregation* and *Instance-aware Query Embedding*. Following the DETR paradigm of set prediction with queries, our framework constructs dedicated *instance queries* from confident detections and propagates them across frames. These queries participate in feature aggregation for the same instances, implicitly integrating partial views from multiple frames into a coherent representation without heuristic post-processing, as shown in Figure 1. Concurrently, *Scene-aware Feature Aggregation* enhances the current frame's features by spatial abstraction from past observations. Notably, *Embodied-DETR* is trained end-to-end from scratch, without relying on pre-trained, frozen single-frame detectors.

**Our contributions are summarized as follows**

1. We introduce *Embodied-Det*, a pioneering benchmark designed for embodied agents that emphasizes egocentric, temporal 3D object detection.

2. We propose *Embodied-DETR*, a novel end-to-end transformer framework for efficient temporal 3D object detection in egocentric views.

3. We design two core modules, *Scene-aware Feature Aggregation* and *Instance-aware Query Embedding*, which model scene-level context and instance-level continuity for effective temporal reasoning.

4. Extensive experiments demonstrate the limitations of existing methods in egocentric settings, while *Embodied-DETR* achieves superior accuracy and temporal consistency, narrowing the gap between research and practical embodied perception.

## 2. Related Work

**Benchmarks for Indoor 3D Object Detection.** Existing benchmarks predominantly provide globally reconstructed indoor scenes. SUN RGB-D (Song et al., 2015), a pioneering dataset, contains a limited set of first-person RGB-D frames but lacks temporal sequences. ScanNet v2 (Dai et al., 2017) and S3DIS (Armeni et al., 2017) offer extensive room scans and complete scene meshes, establishing the standard for scene-level 3D perception. Follow-up works have expanded task scope and data diversity (Johanna Wald, 2019; Mao et al., 2022). However, these benchmarks are designed for offline, global-scene analysis and do not address the temporal, egocentric perception required by embodied agents. EmbodiedScan (Wang et al., 2024) advances the field by introducing first-person view data for perception tasks, yet it does not explicitly model temporal continuity or evaluate temporal detection performance. In contrast, our *Embodied-Det* provides temporal egocentric view data and introduces comprehensive metrics for temporal evaluation, directly targeting the online perception challenges in embodied scenarios.

**3D Object Detection.** Indoor 3D object detection methods can be categorized by their core mechanism. **Voting-based** methods (Qi et al., 2019; Zhang et al., 2020) first generate votes for potential object centers and then refine proposals, effectively handling the absence of inner points. **Expansion-based** methods (Rukhovich et al., 2022; 2023; Wang et al., 2022) employ hierarchical voxel backbones to progressively expand receptive fields for context aggregation. **Transformer-based** methods formulate detection as a set prediction problem, using learnable queries to directly output object sets. This paradigm, exemplified by 3DETR (Misra et al., 2021), GroupFree3D (Liu et al., 2021), and more recent works (Wang et al., 2023; Shen et al., 2024; Kolodiazhnyi et al., 2025; Wang et al., 2025), offers strong flexibility and extensibility. Our work builds upon this query-based Transformer paradigm to construct an end-to-end temporal detector.

**Temporal 3D Object Detection.** Leveraging spatiotemporal continuity can significantly enhance detection efficiency. However, due to the lack of suitable benchmarks, temporal detection in *indoor* scenes remains under-explored, with

most research focusing on autonomous driving. A common strategy employs a single-frame detector followed by heuristic association across frames (Yang et al., 2021; Chen et al., 2022). More advanced approaches fuse multi-frame point cloud features or intermediate representations via attention mechanisms to learn coherent instance representations over time (He et al., 2023; Huang et al., 2024). To the best of our knowledge, *Embodied-DETR* is the first framework designed for *online, end-to-end temporal 3D object detection from egocentric views* in embodied indoor environments.

## 3. Embodied-Det Benchmark

To address the lack of suitable benchmarks for online 3D perception in embodied agents, we introduce ***Embodied-Det***, a benchmark specifically designed for *egocentric* and *temporal* 3D object detection. We present a pipeline to reorganize existing datasets with reconstructed scenes into a sequential, first-person view format, and propose a comprehensive set of evaluation metrics encompassing detection accuracy, scene understanding, and temporal performance.

The core task of *Embodied-Det* is to predict 3D bounding boxes for specified categories *online* within the Field of View (FoV), processing a stream of egocentric RGB-D frames sequentially, as illustrated in Figure 2. We construct the dataset from ScanNet v2 (Dai et al., 2017). Each ScanNet scan is treated as a video sequence, yielding 1,513 sequences in total. After filtering, each sequence contains tens to hundreds of frames, each comprising an RGB-D image, camera pose, and per-frame 3D annotations. In summary, *Embodied-Det* contains over 230k egocentric frames, split into approximately 180k training frames (from 1,213 videos) and 50k test frames (from 300 videos), following ScanNet's original split. To ensure fair comparison with existing work, we adopt the same 18 common indoor object categories as ScanNet, resulting in a dataset with around 500k annotated objects. *More details are provided in appendix A.*

### 3.1. Data Construction

We uniformly sample one frame every ten frames from the original ScanNet videos, so that three consecutive frames in our data correspond to roughly one second in real time. This provides sufficient temporal overlap while avoiding redundant similarity. Frames with invalid camera poses are removed, and all data is rotated to align with ScanNet's canonical coordinate axes. To obtain per-frame 3D bounding box annotations from the global scene mesh, we employ the following pipeline: (1) extract mesh vertices for all instances; (2) project them into the camera coordinate system of the current frame; (3) cull instances outside the current camera frustum; (4) for each remaining instance, compute an overlap score between its mesh vertices and the current frame's point cloud as a visibility measure, discarding in-

stances with a specific threshold $\epsilon_v$. The retained instances form the ground-truth annotations for that frame.

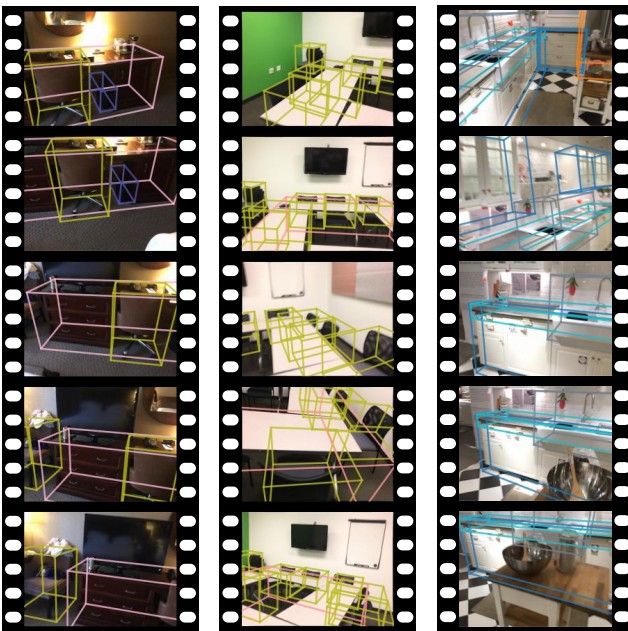

*Figure 2.* Examples from *Embodied-Det*. The benchmark provides sequential egocentric view data (from top to bottom) across diverse scenes (from left to right), highlighting the temporal and partial-observability challenges.

### 3.2. Evaluation Metrics

We employ Average Precision (AP) (Everingham et al., 2010) as the primary detection accuracy metric and introduce novel metrics to quantify *temporal* detection quality in terms of consistency and stability.

**Detection Precision.** We evaluate precision at both the frame and scene levels. Frame-level AP measures instantaneous detection performance from the egocentric perspective across all frames. Scene-level AP aggregates all predictions from a sequence into a unified global coordinate system, then applies an IoU-based clustering to merge predictions for the same instance, reflecting the model's holistic understanding of the scene. Bounding boxes within a cluster are weighted-averaged based on their confidence scores, yielding the cluster box. The confidence score of the fused cluster box is the sum of the constituent boxes' confidence scores. For both levels, we report mean AP (mAP) over the 18 categories at IoU thresholds of 0.25 and 0.5, following the standard ScanNet protocol:

$$\text{mAP} = \frac{1}{|\mathcal{C}|} \sum_{c \in \mathcal{C}} \text{AP}(c). \tag{1}$$

**Temporal Consistency.** This metric assesses the smoothness of bounding boxes for correctly detected instances.

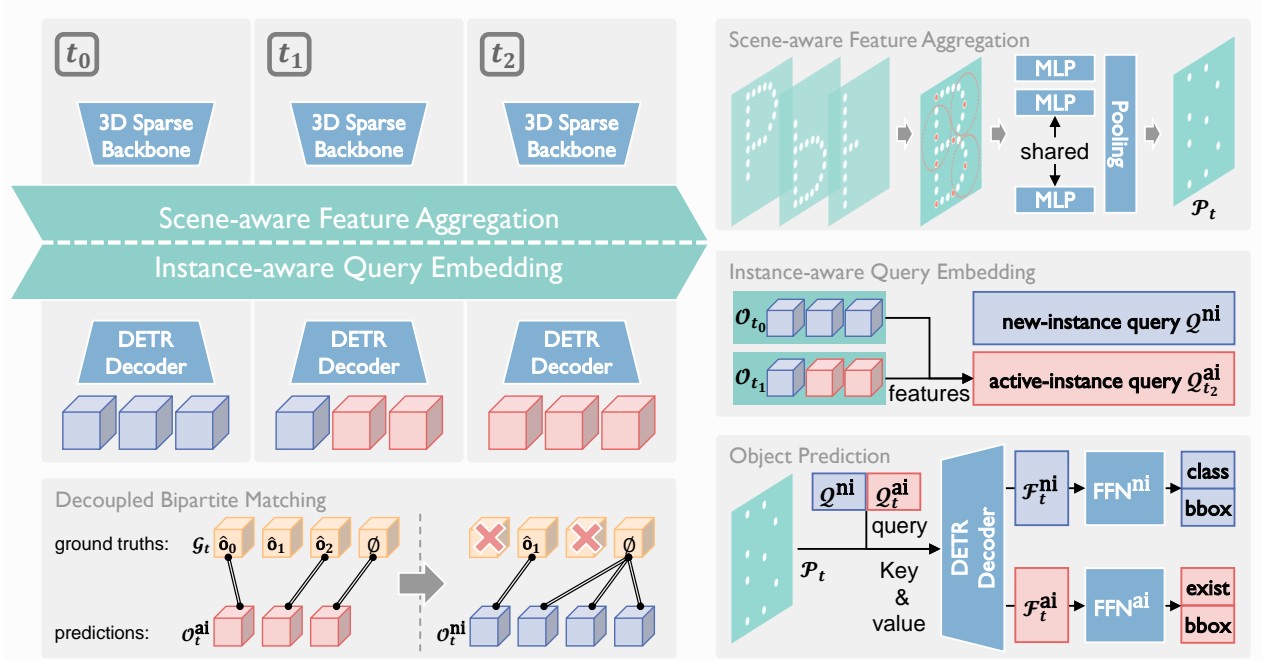

*Figure 3.* Overview of the *Embodied-DETR* framework. The pipeline processes sequential egocentric RGB-D inputs (top-left) and outputs detected objects frame-by-frame. Key components include Scene-aware Feature Aggregation (top-right), Instance-aware Query Embedding (middle-right), and the decoding process (bottom-right). The training-time matching strategy is illustrated at the bottom-left.

For a ground-truth instance $\hat{\mathbf{o}}$ visible in frames $\mathcal{T}_{\hat{\mathbf{o}}} = \{t_1, \ldots, t_{N_{\hat{\mathbf{o}}}}\}$, where $N_{\hat{\mathbf{o}}} = |\mathcal{T}_{\hat{\mathbf{o}}}|$. We define a binary detection sequence $\mathbf{y}_{\hat{\mathbf{o}}} = (y_{\hat{\mathbf{o}}}(t_1), \ldots, y_{\hat{\mathbf{o}}}(t_{N_{\hat{\mathbf{o}}}}))$, where $y_{\hat{\mathbf{o}}}(t_k) = 1$ if $\hat{\mathbf{o}}$ is detected at frame $t_k$, otherwise $y_{\hat{\mathbf{o}}}(t_k) = 0$. For $\hat{\mathbf{o}}$, we consider adjacent frame pairs $(t_k, t_{k+1})$ where it is detected in both frames. Let $\mathcal{K}_{\hat{\mathbf{o}}}$ denote the set of indices of such valid adjacent frame pairs. Let $\mathbf{o}^{\text{loc}}(t) \in \mathbb{R}^3$ and $\mathbf{o}^{\text{size}}(t) \in \mathbb{R}^3$ denote the 3D location and size of a predicted instance $\mathbf{o}$ at frame $t$, and let $\hat{\mathbf{o}}^{\text{loc}}(t)$ and $\hat{\mathbf{o}}^{\text{size}}(t)$ be the corresponding ground-truth values. We define the normalized localization and size errors at frame $t$ as:

$$\boldsymbol{\delta}_{\hat{\mathbf{o}}}^{l}(t) = \frac{\mathbf{o}^{\text{loc}}(t) - \hat{\mathbf{o}}^{\text{loc}}(t)}{\hat{\mathbf{o}}^{\text{size}}(t)}, \quad \boldsymbol{\delta}_{\hat{\mathbf{o}}}^{s}(t) = \log\left(\frac{\mathbf{o}^{\text{size}}(t)}{\hat{\mathbf{o}}^{\text{size}}(t)}\right). \tag{2}$$

The **Average Location Deviation** (ALD) and **Average Size Deviation** (ASD) measure the frame-to-frame jitter for each instance by averaging over all valid adjacent pairs:

$$\text{ALD}(\hat{\mathbf{o}}) = \frac{1}{|\mathcal{K}_{\hat{\mathbf{o}}}|} \sum_{k \in \mathcal{K}_{\hat{\mathbf{o}}}} \left\| \boldsymbol{\delta}_{\hat{\mathbf{o}}}^{l}(t_{k+1}) - \boldsymbol{\delta}_{\hat{\mathbf{o}}}^{l}(t_k) \right\|_2, \tag{3}$$

$$\text{ASD}(\hat{\mathbf{o}}) = \frac{1}{|\mathcal{K}_{\hat{\mathbf{o}}}|} \sum_{k \in \mathcal{K}_{\hat{\mathbf{o}}}} \left\| \boldsymbol{\delta}_{\hat{\mathbf{o}}}^{s}(t_{k+1}) - \boldsymbol{\delta}_{\hat{\mathbf{o}}}^{s}(t_k) \right\|_2. \tag{4}$$

We report the class-balanced means, mALD and mASD, by averaging first over instances within each category and then over all categories. Low deviation values indicate spatially consistent predictions over time, which is crucial

for downstream tasks like motion planning that rely on stable perception outputs.

**Temporal Stability.** This metric evaluates whether an instance, once detected, remains consistently recognized across frames. Using the binary mapping $y_{\hat{\mathbf{o}}}(t)$ for an instance $\hat{\mathbf{o}}$ and treating $\mathbf{y}_{\hat{\mathbf{o}}}$ as a circular sequence to handle continuity, we calculate the longest contiguous subsequence of ones ($L_{\hat{\mathbf{o}}}^{TP}$) and zeros ($L_{\hat{\mathbf{o}}}^{FN}$). The **Longest Continuous Detection Ratio** (CDR) and **Longest Continuous Missing Ratio** (CMR) are:

$$\text{CDR}(\hat{\mathbf{o}}) = \frac{L_{\hat{\mathbf{o}}}^{TP}}{N_{\hat{\mathbf{o}}}}, \qquad \text{CMR}(\hat{\mathbf{o}}) = \frac{L_{\hat{\mathbf{o}}}^{FN}}{N_{\hat{\mathbf{o}}}}. \tag{5}$$

The final metrics, mCDR and mCMR, are obtained via class-balanced averaging. A high mCDR indicates stable detection rather than flicker, while a higher mCMR represents a worse case, indicating the maximum consecutive frames the detector might need to observe before successfully detecting the instance.

## 4. Embodied-DETR

To address the inherent challenges of occlusion and partial observability in egocentric perception, we propose ***Embodied-DETR***, a novel end-to-end framework for temporal 3D object detection, as illustrated in Figure 3. *Embodied-DETR* processes a stream of egocentric RGB-D frames. For each input frame, a 3D sparse backbone extracts point-wise

features. The **Scene-aware Feature Aggregation** module then enhances this representation by spatially aligning and aggregating features from past frames stored in the memory bank, providing broader contextual information. Concurrently, the **Instance-aware Query Embedding** module maintains a dynamic set of queries. Learnable parameter vectors serve as new-instance queries to discover previously unseen objects. Active-instance queries are constructed from features of instances detected in previous frames, enabling the model to integrate multiple partial observations of the same instance over time. A DETR decoder processes these queries alongside the enhanced scene features to produce object representations. These are decoded into semantic categories and bounding boxes, with new-instance queries predicting a full category distribution and active-instance queries predicting a binary existence score for refinement. During training, ground-truth instances are first assigned to the corresponding active-instance queries based on their instance IDs; any remaining unmatched ground truths are assigned to new-instance queries. *Implementation details are provided in appendix B.*

## 4.1. Scene-aware Feature Aggregation

Given an input point cloud for frame $t$, we first use a 3D sparse convolutional backbone (Choy et al., 2019) to extract sparse voxel features. We then sample a set of representative points with both coordinates and features. These points form the local representation of frame $t$ and are added to a global memory bank $\mathcal{M}$ that stores feature points from past frames in a unified coordinate system.

To aggregate context, we retrieve points from $\mathcal{M}$ that lie within the current camera frustum, forming a set of intertemporal points $\mathcal{X} = \{\mathbf{x}_j\}_{j=1}$ with coordinates and features, denoted as $\mathbf{x}_j = (\mathbf{x}_j^{\text{coord}}, \mathbf{x}_j^{\text{feat}})$. From $\mathcal{X}$, we sample $N_s$ seed points $\{\mathbf{s}_i\}_{i=1}^{N_s}$ using Farthest Point Sampling (FPS). For each seed $\mathbf{s}_i$, we gather its neighboring points $\mathcal{N}_r(\mathbf{s}_i)$ from $\mathcal{X}$ within a radius $r$ via a ball query. The enhanced feature for the seed is computed by:

$$\mathbf{p}_i^{\text{feat}} = \max_{\mathbf{x}_j \in \mathcal{N}_r(\mathbf{s}_i)} \phi\left(\left[\mathbf{x}_j^{\text{feat}}, \mathbf{x}_j^{\text{coord}} - \mathbf{s}_i^{\text{coord}}\right]\right), \quad (6)$$

where $\phi(\cdot)$ is a shared MLP, $[\cdot, \cdot]$ denotes concatenation, and $\max$ represents max pooling. The output constitutes a set of scene-aware feature points $\mathcal{P}_t = \{(\mathbf{s}_i^{\text{coord}}, \mathbf{p}_i^{\text{feat}})\}_{i=1}^{N_s}$, which aggregate multi-frame context for more comprehensive perception.

## 4.2. Instance-aware Query Embedding

*Embodied-DETR* employs two distinct types of query vectors. (1) *New-instance queries* $\mathcal{Q}^{\text{ni}} = \{\mathbf{q}_k^{\text{ni}}\}$ are a fixed set of $N_q$ learnable embeddings, identical to standard DETR, responsible for discovering instances appearing for the first

time. (2) *Active-instance queries* $\mathcal{Q}_t^{\text{ai}} = \{\mathbf{q}_k^{\text{ai}}\}$ are dynamically constructed from instance features stored in the memory bank $\mathcal{M}$, each corresponding to a specific physical instance that has been detected over time. The complete query set for frame $t$ is the union $\mathcal{Q}_t = \mathcal{Q}^{\text{ni}} \cup \mathcal{Q}_t^{\text{ai}}$.

At the first frame ($t = 0$), $\mathcal{M}$ is empty, so $\mathcal{Q}_0 = \mathcal{Q}^{\text{ni}}$. Detected reliable instances are then added to $\mathcal{M}$, including their features and bounding boxes. In subsequent frames, for each instance in $\mathcal{M}$ that falls within the current FoV, an active-instance query $\mathbf{q}_k^{\text{ai}}$ is constructed from its latest feature representation. If this query successfully detects the instance, the instance's state in $\mathcal{M}$ is updated, including the more complete representation obtained after the interaction between $\mathbf{q}_k^{\text{ai}}$ and the latest observation. If $\mathbf{q}_k^{\text{ai}}$ fails to detect the corresponding instance for consecutive frames, it is removed from $\mathcal{M}$. This mechanism encourages modeling the complete representation of instances from multiple observations end-to-end, thereby gradually achieving more refined detection.

## 4.3. Object Prediction

The prediction module takes the scene-aware feature points $\mathcal{P}_t$ and the queries $\mathcal{Q}_t$ as input. A DETR-style transformer decoder performs cross-attention between queries and features, producing an output representation for each query:

$$\mathcal{F}_t = \{\mathbf{f}_m\}_{m=1}^{|\mathcal{Q}_t|} = \text{Decoder}(\mathcal{Q}_t, \mathcal{P}_t), \quad (7)$$

where $\mathbf{f}_m \in \mathbb{R}^D$ with dimensions $D$ denotes the decoded feature associated with one query. These representations are split according to their query type: $\mathcal{F}_t^{\text{ni}}$ from $\mathcal{Q}^{\text{ni}}$ and $\mathcal{F}_t^{\text{ai}}$ from $\mathcal{Q}_t^{\text{ai}}$. Two separate perceptrons $\text{FFN}^{\text{ni}}$ and $\text{FFN}^{\text{ai}}$ process them to obtain the objects $\mathcal{O}_t$. For a new-instance representation $\mathbf{f}_i^{\text{ni}}$, $\text{FFN}^{\text{ni}}$ predicts a categorical distribution $\mathbf{c}_i$ over all classes and a bounding box $\mathbf{b}_i$. For an active-instance representation $\mathbf{f}_j^{\text{ai}}$, $\text{FFN}^{\text{ai}}$ predicts a binary existence score $s_j \in [0, 1]$ and a bounding box $\mathbf{b}_j$. For a prediction $\mathbf{o}_k \in \mathcal{O}_t$ with a confidence higher than the threshold $\epsilon_r$, its representation $\mathbf{f}_k$ and bounding box $\mathbf{b}_k$ will be added to or used to update $\mathcal{M}$.

## 4.4. Training

We introduce a set of unified end-to-end training strategies that jointly optimize all parameters of *Embodied-DETR* from scratch, enabling coherent learning of both detector and temporal modeling.

**Decoupled Bipartite Matching.** At each frame $t$, let $\mathcal{G}_t = \{\hat{\mathbf{o}}_m\}$ denote the set of ground-truth instances, each associated with a unique instance ID. The matching process is decoupled based on query type. For a ground-truth $\hat{\mathbf{o}}_m$ with instance ID $k$, if an active-instance query $\mathbf{q}_k^{\text{ai}} \in \mathcal{Q}_t^{\text{ai}}$ exists, $\hat{\mathbf{o}}_m$ is directly assigned to it. This ensures consistent supervision for persistent instances across frames. All

*Table 1.* Comparison on the *Embodied-Det* test split set. The best results are shown in **bold**.

| Method | Reference | Frame-Level | | Scene-Level | | Consistency | | Stability | | Latency↓ |
|---|---|---|---|---|---|---|---|---|---|---|
| | | mAP$_{25}$↑ | mAP$_{50}$↑ | mAP$_{25}$↑ | mAP$_{50}$↑ | mALD↓ | mASD↓ | mCDR↑ | mCMR↓ | |
| VoteNet | ICCV '19 | 45.24 | 25.50 | 53.88 | 36.59 | 0.1192 | 1.2211 | 28.43 | 51.78 | 71 ms |
| H3DNet | ECCV '20 | 49.61 | 28.70 | 58.26 | 40.51 | 0.1162 | 0.9052 | 31.78 | 48.90 | 184 ms |
| GroupFree3D | ICCV '21 | 40.56 | 22.84 | 52.39 | 34.45 | 0.0990 | 0.9106 | 24.48 | 58.03 | 110 ms |
| FCAF3D | ECCV '22 | 51.89 | 34.80 | 58.98 | 44.68 | 0.0852 | 0.9972 | 39.21 | 42.72 | 132 ms |
| TR3D | ICIP '23 | 54.99 | 36.07 | 61.46 | 45.78 | 0.0966 | 0.9190 | 44.26 | **37.21** | **55** ms |
| V-DETR | ICLR '24 | 57.35 | 39.14 | 66.88 | 50.28 | 0.0675 | 0.8266 | 37.74 | 45.89 | 156 ms |
| UniDet3D | AAAI '25 | 45.15 | 22.67 | 55.07 | 35.91 | 0.0993 | 0.7537 | 28.61 | 51.35 | 62 ms |
| *Embodied-DETR* (**Ours**) | | **59.29** | **42.90** | **69.57** | **52.28** | **0.0271** | **0.6895** | **54.23** | 39.37 | 112 ms |

remaining unmatched ground-truth instances (newly appearing or previously missed instances) are then matched to the new-instance queries $\mathcal{Q}^{\text{ni}}$ using the standard Hungarian matching algorithm, with a cost function combining classification and box regression terms. This strategy dedicates $\mathcal{Q}^{\text{ai}}$ to refining known instances and $\mathcal{Q}^{\text{ni}}$ to discovering new ones. All queries that do not match any ground truths are assigned to the $\emptyset$ (no object).

**Loss Function.** For each matched (positive only) prediction, we compute a bounding box regression loss $\mathcal{L}_{\text{reg}}$ and a 3D IoU loss $\mathcal{L}_{\text{iou}}$. For all predictions (positive and negative), we employ a Focal loss (Lin et al., 2017) as the classification loss $\mathcal{L}_{\text{cls}}$. The supervision signal differs by query type: predictions from matched new-instance queries are supervised with one-hot category labels, while matched active-instance queries receive the binary label 1 for existence. Unmatched (negative only) predictions are supervised with the background label 0 for classification only. The total loss for a training sequence $\mathcal{S}$ aggregates losses over all frames and predictions, normalized by the total number of ground-truth objects in the sequence:

$$\mathcal{L} = \frac{1}{\sum_{t \in \mathcal{S}} |\mathcal{G}_t|} \sum_{t \in \mathcal{S}} \sum_{\mathbf{o} \in \mathcal{O}_t} \left( \mathcal{L}_{\text{reg}} + \mathcal{L}_{\text{iou}} + \mathcal{L}_{\text{cls}} \right), \quad (8)$$

**Curriculum Learning.** Training directly on long sequences can reduce data stochasticity and hinder optimization. We therefore employ a curriculum learning strategy. Initially, mini-batches are composed of many short clips sampled randomly from different scenes, allowing the model to quickly master single-frame detection. As training progresses, the clip length is gradually increased, guiding the model to learn longer-term temporal dependencies and instance evolution.

**Spatial Sampling.** To foster efficient association learning, we construct training clips by clustering frames based on the spatial overlap of their observed regions, rather than solely on temporal order. Frames within a cluster form a training clip, guaranteeing partial overlap between each other. Cluster centers are sampled by FPS to obtain differentiated clusters. This spatial data sampling, combined with random

clustering, increases the diversity of training sequences and encourages learning of long-range context associations.

**Adaptive Threshold.** The memory bank update hinges on a confidence threshold $\epsilon_r$ for adding or refreshing instances. Instead of a fixed heuristic value, we adopt a self-adaptive mechanism. During training, we record the confidence and positivity of all instance predictions within one epoch. At the end of each epoch, we select the confidence threshold that maximizes the F1 score on the training set and use it for memory bank management in the next epoch. This allows the threshold to co-evolve with the model's detection capability. Empirically, it converges to a stable value and rarely changes in later stages.

**Group Query.** Inspired by Group DETR (Chen et al., 2023), we replicate the set of active-instance queries into $G$ isolated groups to stabilize training. Only the predictions from the first group are used to update the memory bank, avoiding the complexity of maintaining multiple parallel memory states. All $G$ groups contribute to the loss computation during training, providing richer supervision. During inference, only the first group is active, ensuring efficiency and consistency.

## 5. Experiments

### 5.1. Settings

All experiments are conducted on our *Embodied-Det* benchmark using the data splits and evaluation metrics defined in Section 3. Our implementation of *Embodied-DETR* is built upon the MMDetection3D framework (Contributors, 2020). We first compare *Embodied-DETR* against a range of widely-used and *state-of-the-art* (SOTA) indoor 3D object detectors. All models are trained for 20 epochs on the full training set, and the final checkpoint is evaluated on the test set. We also verify the impact of different input perspectives on detection performance. Subsequently, to analyze the contribution of each proposed component, we perform comprehensive ablation studies on the core temporal modules and training strategies. For efficiency, all ablated variants are trained on a quarter (1/4) of the training set but eval-

*Table 2.* Comparison of per-category frame-level $AP_{50}$ on the *Embodied-Det* test split set. The best results are shown in **bold**.

| Method | Cab | Bed | Chair | Sofa | Table | Door | Wind | Bsh | Pic | Ctr | Desk | Curt | Frdg | ShCurt | Toilet | Sink | Tub | Other | mean |
|---|---|---|---|---|---|---|---|---|---|---|---|---|---|---|---|---|---|---|---|
| VoteNet | 8.02 | 66.48 | 46.65 | 60.23 | 27.72 | 10.45 | 2.68 | 20.48 | 0.41 | 1.26 | 28.89 | 3.22 | 15.11 | 4.90 | 82.66 | 20.22 | 44.94 | 14.74 | 25.50 |
| H3DNet | 13.13 | 66.95 | 52.27 | 65.47 | 33.75 | 10.48 | 5.85 | 16.07 | 3.44 | 0.55 | 34.86 | 2.10 | 17.42 | 6.89 | 85.86 | 28.01 | 54.38 | 19.03 | 28.70 |
| GroupFree3D | 3.06 | 56.08 | 35.12 | 54.05 | 22.81 | 9.60 | 2.00 | 8.49 | 2.07 | 0.66 | 25.78 | 1.44 | 14.00 | 4.33 | 81.96 | 26.11 | 47.81 | 15.73 | 22.84 |
| FCAF3D | 22.30 | 69.70 | 65.15 | **67.49** | 41.98 | 21.13 | 7.68 | 20.77 | 10.81 | 5.32 | 37.83 | 9.72 | 20.75 | 8.53 | 87.62 | 33.14 | 61.41 | 35.02 | 34.80 |
| TR3D | 23.37 | 70.47 | 65.74 | 66.34 | 43.19 | 20.76 | 7.73 | 21.81 | 14.11 | 6.18 | **44.47** | 6.24 | 22.59 | 14.42 | 89.01 | **35.77** | 64.01 | 33.06 | 36.07 |
| V-DETR | 24.76 | 72.11 | 66.70 | 63.68 | 44.49 | 27.96 | 14.01 | 24.45 | 19.17 | 7.11 | 44.38 | 19.19 | 22.38 | 25.53 | 90.07 | 34.64 | 67.46 | 36.42 | 39.14 |
| UniDet3D | 13.84 | 34.64 | 41.29 | 41.52 | 32.74 | 4.67 | 2.27 | 11.61 | 11.31 | 0.87 | 26.56 | 1.90 | 5.96 | 0.09 | 75.38 | 30.41 | 44.42 | 28.60 | 22.67 |
| **Ours** | **27.22** | **73.82** | **68.90** | 67.30 | **50.59** | **31.15** | **17.73** | **28.32** | **19.53** | **16.05** | 44.22 | **29.34** | **29.02** | **30.20** | **93.08** | 33.13 | **72.37** | **40.17** | **42.90** |

*Table 3.* Performance change compared to global view detection.

| Method | Scene-Level | | ScanNet v2 | |
|---|---|---|---|---|
| | $mAP_{25}\uparrow$ | $mAP_{50}\uparrow$ | $mAP_{25}\uparrow$ | $mAP_{50}\uparrow$ |
| VoteNet | 53.9 (-4.7) | 36.6 (+3.1) | 58.6 | 33.5 |
| H3DNet | 58.3 (-8.9) | 40.5 (-7.6) | 67.2 | 48.1 |
| GroupFree3D | 52.4 (-16.7) | 34.5 (-18.3) | 69.1 | 52.8 |
| FCAF3D | 59.0 (-12.5) | 44.7 (-12.6) | 71.5 | 57.3 |
| TR3D | 61.5 (-11.4) | 45.8 (-13.5) | 72.9 | 59.3 |
| V-DETR | 66.9 (-10.5) | 50.3 (-14.7) | 77.4 | 65.0 |
| UniDet3D | 55.1 (-22.4) | 35.9 (-30.2) | 77.5 | 66.1 |

*Table 4.* Effect of the temporal module.

| SFA | IQE | Frame-Level | | Consistency | |
|---|---|---|---|---|---|
| | | $mAP_{25}\uparrow$ | $mAP_{50}\uparrow$ | mALD↓ | mASD↓ |
| | | 45.67 | 28.48 | 0.0880 | 0.9775 |
| ✓ | | 47.20 | 31.50 | 0.0795 | 0.8856 |
| | ✓ | 48.87 | 33.89 | 0.0434 | 0.7455 |
| ✓ | ✓ | **50.36** | **35.81** | **0.0347** | **0.7323** |

uated on the full test set, following Li et al. (2022). This protocol maintains a fair comparison while significantly reducing computational cost. *More implementation details are provided in appendix C.*

## 5.2. Main Results

We evaluate *Embodied-DETR* against a suite of representative and SOTA indoor 3D detectors: VoteNet (Qi et al., 2019), H3DNet (Zhang et al., 2020), GroupFree3D (Liu et al., 2021), FCAF3D (Rukhovich et al., 2022), TR3D (Rukhovich et al., 2023), V-DETR (Shen et al., 2024), and UniDet3D (Kolodiazhnyi et al., 2025). All methods are trained and tested on *Embodied-Det*, with comprehensive results presented in Table 1.

**Performance of *Embodied-DETR*.** By deep modeling temporal continuity, *Embodied-DETR* achieves superior performance across nearly all metrics. It achieves the best frame-level and scene-level mAP, with notable leads of +3.76 and +2.00 in $mAP_{50}$, respectively, over the strongest baseline V-DETR. More importantly, *Embodied-DETR* establishes a significant advantage in temporal metrics. It substantially reduces temporal jitter (mALD & mASD) and improves temporal stability (mCDR) by a large margin (*e.g.*, +16.49 over V-DETR), demonstrating its ability to produce consistent instance representations over time. The only relative weakness is a slightly higher mCMR, suggesting that detecting brand-new instances the moment they enter the FoV remains a challenging frontier. Meanwhile, *Embodied-DETR* achieves the highest $AP_{50}$ for the majority of the

18 categories, as detailed in Table 2. It shows particular improvement in detecting common yet often occluded furniture, such as chairs, tables, and curtains, validating the benefit of temporal aggregation for handling partial views.

**Performance Change from Global to Egocentric View.** The overall ranking of methods on our benchmark exhibits a partial correlation with their performance on ScanNet, which provides complete scene meshes. However, as shown in Table 3, almost all methods suffer substantial performance degradation in the egocentric setting. For instance, high-performing methods V-DETR and UniDet3D experience a relative decline of over 10 points in $mAP_{50}$. This underscores the distinct challenge posed by online, first-person perception and reveals a critical gap in the generalization capability of current global-view detectors. The degradation is particularly pronounced for UniDet3D, whose performance might heavily rely on pre-computed superpoint segmentation, which is effective on clean, reconstructed meshes but fragile under the noise of RGB-D point clouds. This sensitivity highlights how our benchmark exposes vulnerabilities related to sensor noise and algorithmic assumptions that are often masked in traditional offline benchmarks.

**Efficiency.** Our framework maintains practical efficiency, processing each frame in 112 ms on average, which is faster than the previous SOTA detector V-DETR (156 ms) and enables near-real-time operation.

## 5.3. Ablation Studies

**Effect of Temporal Modules.** To systematically evaluate our proposed components, we conduct ablation studies using the following notation. The variant with all temporal

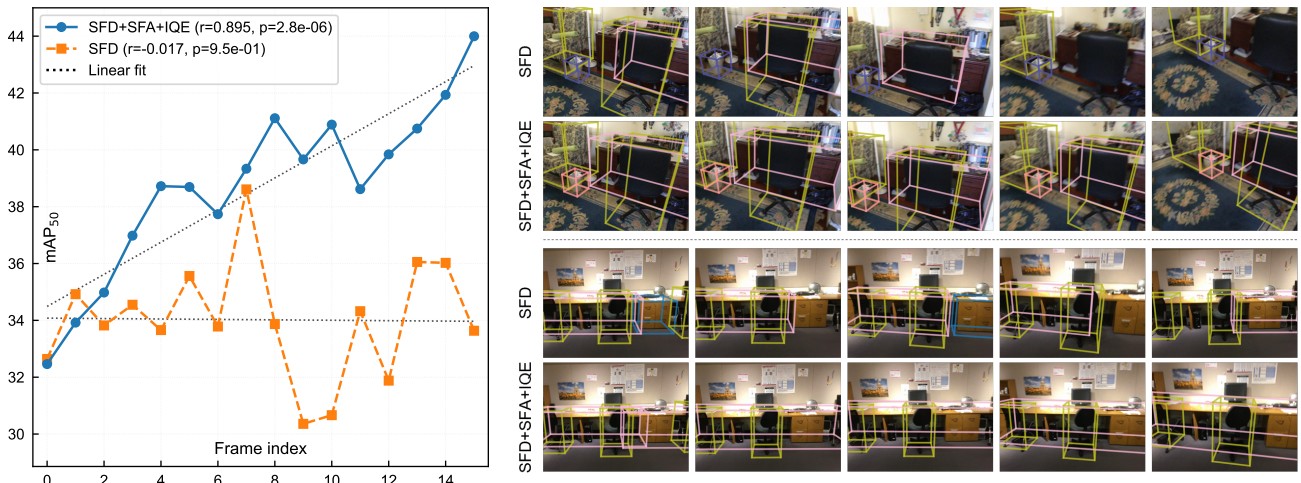

*Figure 4.* (left) Per-frame performance over time, and (right) qualitative comparison of the temporal module.

modules removed, retaining only single-frame detection, is denoted **SFD** (Single-Frame Detector) and closely resembles the original V-DETR architecture. Our core temporal modules, *Scene-aware Feature Aggregation* and *Instance-aware Query Embedding*, are abbreviated as **SFA** and **IQE**, respectively. We first validate the contributions of the SFA and IQE modules by incrementally adding them to the SFD baseline. As shown in Table 4, each module individually improves performance, and their combination yields the best results across all metrics. This demonstrates that both scene-level and instance-level context aggregation are crucial for accurate temporal detection. Moreover, the instance-level modeling (IQE) contributes more substantially to the performance gain, bringing an additional +2.39 $mAP_{50}$ over the scene-only variant. To further analyze the temporal behavior, we compute per-frame mAP by aligning frames across all test sequences by their frame index. As illustrated in Figure 4 (left), the performance of the full model (SFD+SFA+IQE) shows a strong positive correlation with the frame index (Pearson $r = 0.895$, $p < 0.001$), indicating that the model effectively accumulates information over time. In contrast, the vanilla SFD baseline shows no significant temporal correlation ($r = -0.017$, $p > 0.950$). This quantitative evidence confirms that our temporal modules enable the detector to leverage historical observations. Qualitative results (Figure 4 right) further illustrate the advantages. The full model produces consistent detections with stable bounding boxes across frames, effectively mitigating flickering and missed detections caused by occlusion or viewpoint changes. The baseline, however, suffers from frequent failures and inconsistencies.

**Effect of Training Strategies.** We next ablate the proposed training strategies, starting from a baseline that includes the temporal modules but uses none of our specialized strategies. As reported in Table 5, this baseline performs poorly, under-

*Table 5.* Effect of the training strategy.

| Improvements | Frame-Level | | Consistency | |
| --- | --- | --- | --- | --- |
| | $mAP_{25}\uparrow$ | $mAP_{50}\uparrow$ | $mALD\downarrow$ | $mASD\downarrow$ |
| baseline | 43.80 | 29.10 | 0.0743 | 0.8603 |
| + curriculum learning | 47.87 | 32.14 | 0.0579 | 0.8828 |
| + spatial sampling | 48.94 | 33.70 | 0.0451 | 0.8070 |
| + adaptive threshold | 49.71 | 34.49 | 0.0501 | 0.7592 |
| + group query | **50.36** | **35.81** | **0.0347** | **0.7323** |

scoring the need for tailored training for temporal detection. Introducing *curriculum learning* brings a significant performance jump (*e.g.*, +4.07 $mAP_{25}$ and +3.04 $mAP_{50}$ over the baseline). This strategy allows the model to first master single-frame detection before learning long-term dependencies. Replacing simple temporal sampling with *spatial sampling* further improves $mAP_{50}$ by 1.56 points. By constructing sequences based on spatial overlap, this strategy increases the diversity of training clips and promotes learning of long-range instance associations. The *adaptive threshold* mechanism provides another clear gain, outperforming a fixed threshold. It dynamically adjusts the confidence threshold based on the model's current performance, fostering more stable memory bank updates throughout training. Finally, incorporating the *group query* mechanism yields a notable improvement, particularly on the stricter $mAP_{50}$ metric (+1.32 points). By replicating active-instance queries into multiple groups, it stabilizes training and enriches supervision without complicating inference.

## 6. Conclusion

In this work, we address the problem of egocentric 3D object detection, a critical capability for embodied agents operating under sequential, partial, and occluded observations.

To bridge the gap with realistic operational scenarios, we introduce *Embodied-Det*, a benchmark tailored for evaluating temporal 3D detection from a first-person perspective. Furthermore, we propose *Embodied-DETR*, an end-to-end detection framework that deeply models spatiotemporal continuity through its core modules: Scene-aware Feature Aggregation and Instance-aware Query Embedding. Experiments demonstrate that our method outperforms existing methods, validating the importance of dedicated temporal modeling for efficient embodied perception.

**Limitations and Future Work.** Our study is currently limited to static environments, as indoor datasets typically focus only on static scenes. A key future direction is to extend the benchmark and method to handle dynamic objects, a common challenge in real-world deployments.

## Impact Statement

This paper presents work aimed at advancing the field of Machine Learning. There are many potential societal consequences of our work, none of which we feel must be specifically highlighted here.

## Acknowledgements

This work was supported by National Natural Science Foundation of China (No.62576107), and the Science and Technology Commission of Shanghai Municipality (No.24511103300), and the Science and Technology Commission of Shanghai Municipality (No.25DZ2200800), and the Science and Technology Commission of Shanghai Municipality (No.24511104200).

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

In this technical appendix, we introduce benchmark details in Section A, implementation details in Section B, and experiment setting details Section C.

## A. Benchmark Details

### A.1. Per-frame Object Annotation and Visible Threshold Selection

To construct a benchmark suitable for online detection in embodied scenarios, we need to generate per-frame object-detection annotations from complete existing scenes, as shown in Figure 5. The key challenge lies in determining whether an object is sufficiently visible from a specific egocentric viewpoint to warrant annotation in that frame. We adopt the ICP (Iterative Closest Point) fitness as a quantitative measure of visibility and empirically determine an appropriate visibility threshold.

For each instance from the reconstructed scene mesh, we register all its mesh vertices (source points $\mathcal{P}_{\text{ins}}$) against the point cloud back-projected from the current frame's RGB-D image (target points $\mathcal{P}_{\text{frame}}$). The steps are as follows:

1. **Coordinate Transformation**: Transform the object mesh vertices $\mathcal{P}_{\text{ins}}$ from the global scene coordinate system to the camera coordinate system of the current frame.

2. **View Frustum Culling**: Discard vertices that fall completely outside the current camera's view frustum after transformation.

3. **Nearest Point Matching**: Using the Open3D library (Zhou et al., 2018), we perform a single nearest-neighbor search with max distance $d_{max} = 0.1\,\text{m}$. No iterative optimization is performed, and the goal is to evaluate the coverage between the two point sets efficiently.

4. **Fitness Calculation**: The ICP fitness is defined as the proportion of source points $\mathcal{P}_{\text{ins}}$ for which a corresponding point exists in the target point cloud $\mathcal{P}_{\text{frame}}$ within the distance $d_{max}$:

$$\text{fitness} = \frac{|\{v \in \mathcal{P}_{\text{ins}} \mid \exists p \in \mathcal{P}_{\text{frame}}, \|v - p\|_2 < d_{max}\}|}{|\mathcal{P}_{\text{ins}}|} \quad (9)$$

The value ranges in [0, 1]. A fitness of 1 indicates the entire surface of the instance is theoretically observed in the current frame's point cloud, while 0 indicates no observable surface.

Selecting an appropriate fitness threshold $\epsilon_v$ to determine object visibility required careful empirical analysis. We evaluated a set of candidate thresholds $\epsilon_v \in$

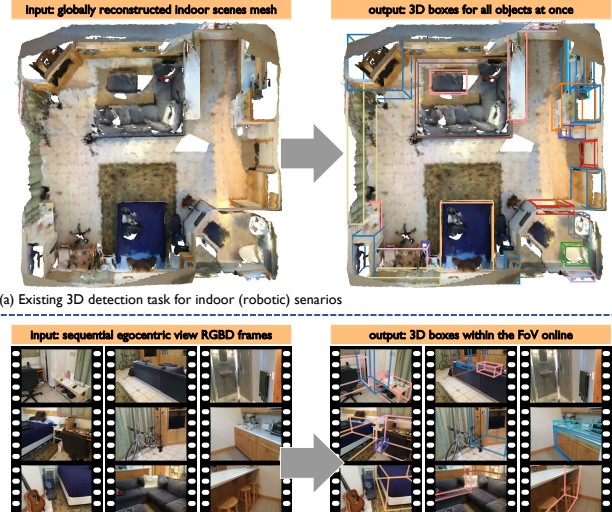

(a) Existing 3D detection task for indoor (robotic) senarios

(b) Our proposed embodied 3D detection task

*Figure 5.* Difference between the existing robotic indoor 3D object detection task (a) and ours (b). Existing benchmarks (a) predominantly provide globally reconstructed indoor scenes meshes. They are designed for offline, global-scene understanding and do not address the temporal, egocentric perception required by embodied agents. In contrast, our Embodied-Det (b) provides temporal RGBD frames in egocentric views and introduces comprehensive metrics for temporal evaluation, requiring handling a stream of egocentric RGB-D frames sequentially.

$\{0.01, 0.25, 0.5, 0.75, 0.99\}$ by manually inspecting the retained object bounding boxes overlaid on RGB and point cloud data across randomly sampled frames. An overly low threshold (e.g., 0.01) resulted in excessive false positives when objects were barely discernible, introducing significant label noise. Conversely, an excessively high threshold (*e.g.*, 0.75 or 0.99) resulted in extremely sparse annotations, as most objects are only partially visible from an egocentric view due to occlusion and limited FoV. Moderate thresholds effectively retained objects with a substantially visible portion, forming a recognizable local structure. Figure 6 illustrates the effect of different $\epsilon_v$ values on the same frames. Balancing annotation purity against sufficient positive sample coverage for learning, we selected $\epsilon_v = 0.25$ as the final visibility filter. This choice ensures that annotated objects possess clear visual evidence of a significant part, aligning with the partial observability inherent in embodied online detection.

### A.2. Scene-level Prediction Aggregation Strategy

To aggregate per-frame predictions into scene-level predictions mentioned in Section 3.2, we propose an overlap-based clustering and fusion strategy. After transforming all per-frame predictions into the global scene coordinate system, we obtain a large set of overlapping bounding boxes, which

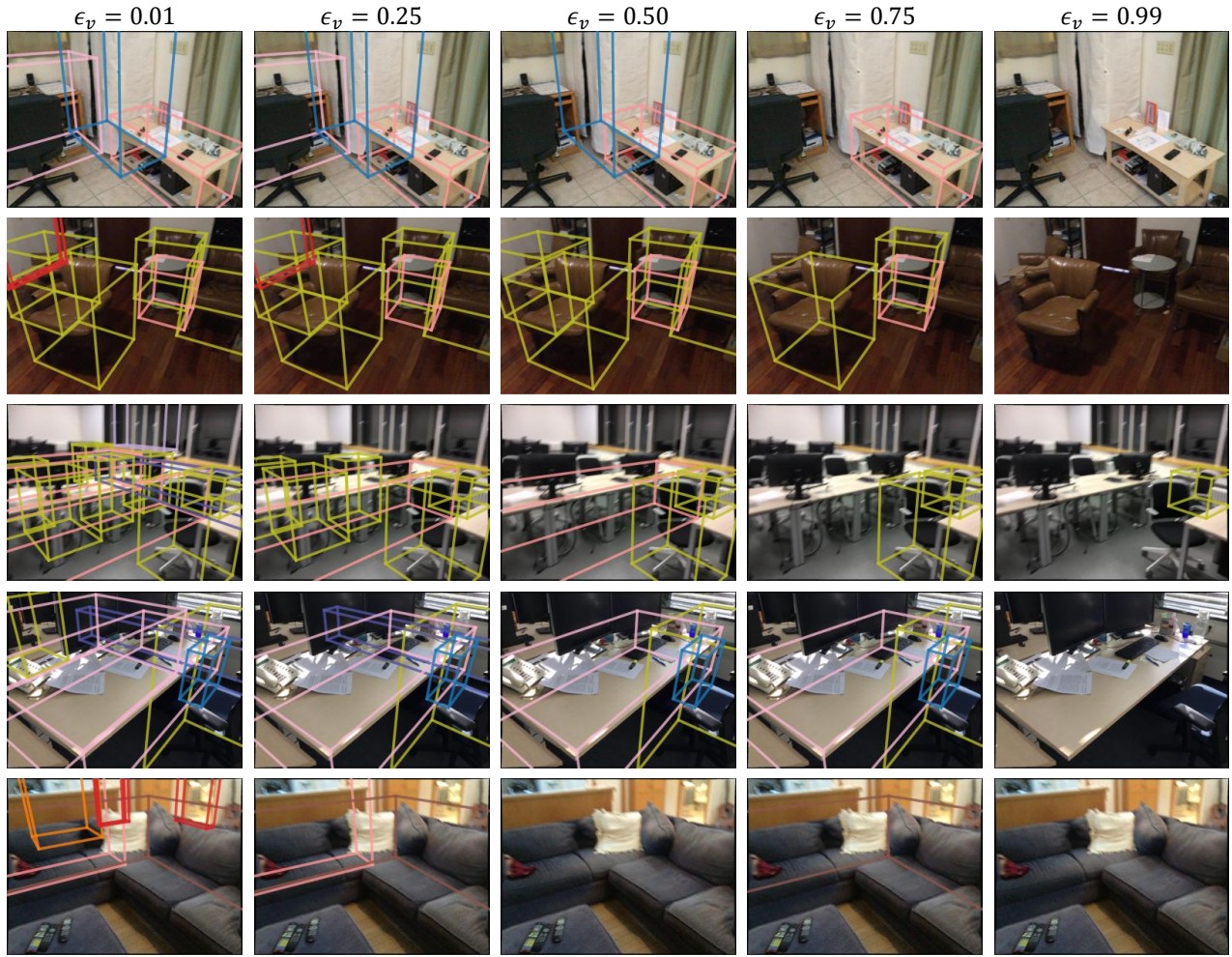

$\epsilon_v = 0.01$  $\epsilon_v = 0.25$  $\epsilon_v = 0.50$  $\epsilon_v = 0.75$  $\epsilon_v = 0.99$

*Figure 6.* Annotations under different $\epsilon_v$.

typically correspond to the same instance observed from different viewpoints. To remove redundancy while preserving the model's understanding of the scene, we employ a heuristic merging procedure. First, we sort all predicted boxes in descending order of confidence. Then, we iterate through the sorted list. For each box that has not yet been assigned to a cluster, we designate it as a seed and group all unassigned boxes with an IoU greater than $0.5$ (the same threshold used for positive matching in our benchmark) with it into the same cluster. Subsequently, for each cluster, we compute a fused bounding box by averaging the box parameters of its members, weighted by their confidence. The confidence score of the cluster box is set as the sum of the confidence scores of all boxes in the cluster. The influence of different confidence aggregation strategies is shown in Figure 7. Finally, we retain only these fused boxes as the model's scene-level understanding derived from the sequence of egocentric views. This process effectively consolidates multiple observations of the same object into a single, high-confidence prediction. In addition, it can also

suppress the random noise that appears in each frame prediction. The randomness of the noise means the resulting scene-level noise boxes usually have low confidence scores, making them more easily suppressed by high-confidence positive cluster boxes.

### A.3. Class Distribution and Imbalance

The $18$ object categories follow the original ScanNet v2 specification. Figure 8 shows the distribution of annotated object instances across categories in the training set. While a natural long-tail distribution exists (e.g., 'chair' and 'table' are abundant, 'refrigerator' and 'toilet' are less frequent), we intentionally preserve this imbalance as it reflects the real-world composition of indoor environments. All reported mean metrics (mAP, mCDR, etc.) are calculated class-averaged to prevent performance from being dominated by frequent categories. In addition, we present the AP across different categories in Table 2. Due to space limitations, the original table only marked the abbreviations of the

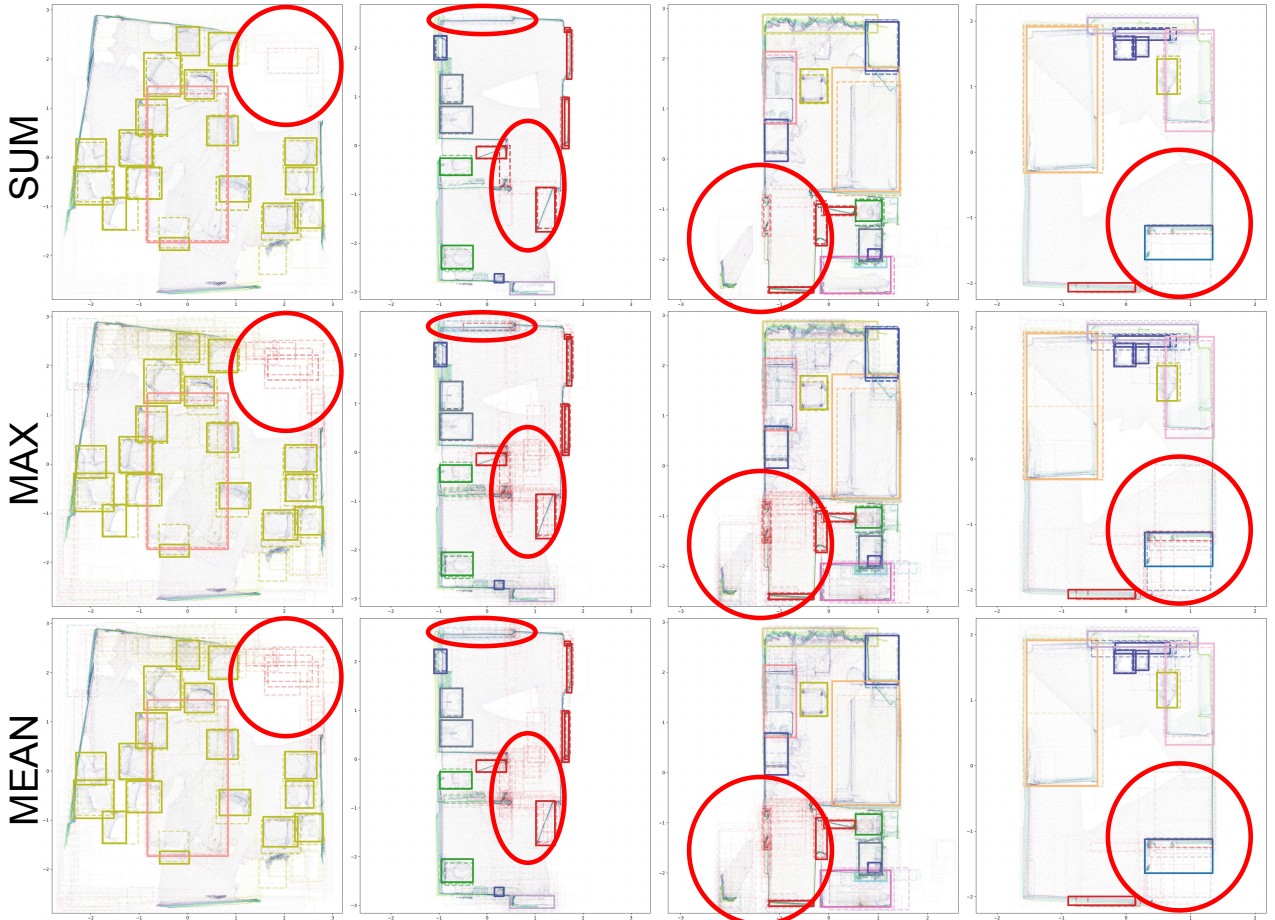

*Figure 7.* Scene-level predictions obtained from the same per-frame detections under different confidence aggregation strategies. The top row shows `sum`, the middle row `max`, and the bottom row `mean`. Solid bounding boxes denote ground truth, while dashed boxes indicate predictions. Different colors correspond to different categories. The opacity (alpha) of each predicted box reflects its confidence, where lower confidence corresponds to lower opacity. Red circles highlight regions with notable differences across aggregation strategies. Compared to max and mean, sum suppresses low-frequency false positives by reinforcing high-frequency true positives, leading to improved robustness against noise.

category names. Their full names are as follows: 'cabinet', 'bed', 'chair', 'sofa', 'table', 'door', 'window', 'bookshelf', 'picture', 'counter', 'desk', 'curtain', 'refrigerator', 'shower curtain', 'toilet', 'sink', 'bathtub', 'otherfurniture'.

### A.4. Comparison to Global View Detection

Table 3 in the main text quantifies the performance change of existing methods when switching from global, complete scene input to our sequential egocentric input. To further illustrate the nature of this challenge, Figure 9 provides a qualitative side-by-side comparison. It shows how a SOTA global-view detector (trained on ScanNet) successfully detects objects when provided the whole scene mesh, but suffers from severe misses and fragmented predictions when its input is restricted to the accumulated point cloud from a sequential egocentric trajectory. This visually underscores the benchmark's focus on the *online partial observability*

challenge.

## B. Implementation Details

### B.1. Network

**3D Sparse Backbone.** Our model employs a 3D sparse convolutional backbone, following the V-DETR design (Shen et al., 2024). It consists of a 3D sparse variant of ResNet34 followed by a Feature Pyramid Network (FPN) built with transposed convolutions. The use of transposed convolutions in the FPN ensures efficient upsampling without inflating the number of voxels. This backbone outputs 3D sparse voxel features with a channel dimension of 256 and a downsampling stride of 4 relative to the original voxel resolution.

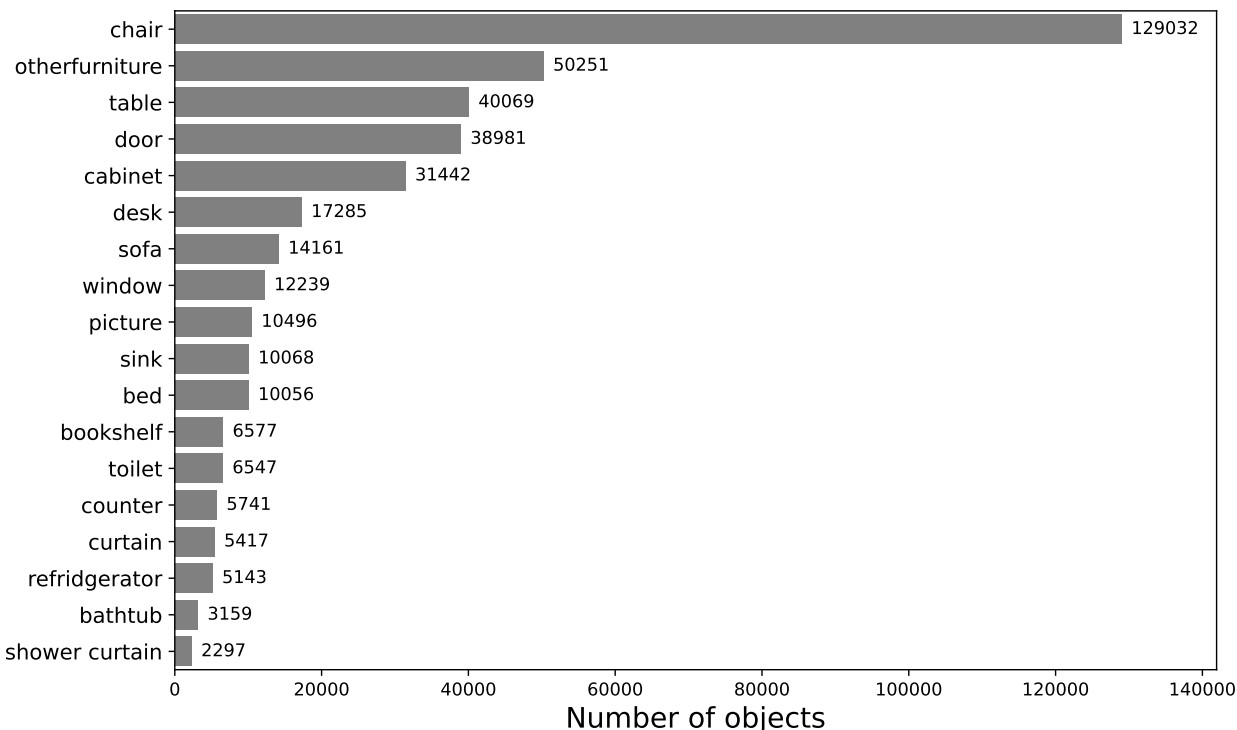

*Figure 8.* Class distribution in training set.

**Query Construction.** For new-instance queries ($\mathcal{Q}^{\mathrm{ni}}$), we utilize a set of 32 learnable parameter vectors. This is significantly smaller than the typical 256+ queries used in standard DETR frameworks for 3D detection, reducing computational overhead while being sufficient for discovering new objects in the local egocentric view. Active-instance queries ($\mathcal{Q}_t^{\mathrm{ai}}$) are a dynamic set of queries constructed directly from the instance features stored in the memory bank. Their number depends on how many active instances are currently within the FoV and have not been removed due to prolonged absence.

**Key & Value Feature Generation.** To generate the key and value features for the transformer decoder, we first sample 256 seed points FPS from the current frame's backbone output features. We concatenate these with another 256 points sampled via FPS from the intertemporal features retrieved from the memory bank (those within the current FoV). This combined set of 512 points serves as the seed points for set abstraction. We perform a multi-scale set abstraction operation on these seeds using the memory bank features. For each seed, we gather neighboring points within two ball query radii ($0.4\,\mathrm{m}$ and $0.8\,\mathrm{m}$). The features of neighbors are processed by a shared MLP and aggregated via max-pooling. The resulting *scene-aware intertemporal features* are passed through three standard transformer encoder layers to produce the final key & value features for the decoder.

**DETR Decoder.** We adopt the transformer decoder from V-DETR, which incorporates 3DV-RPE to enhance spatial reasoning. A key requirement of this decoder is that each query must be associated with a reference bounding box to guide the cross-attention mechanism.

- For new-instance queries, we predict a set of coarse bounding boxes directly from the key features via a lightweight MLP head. We select the top-32 boxes with the highest objectness scores as the reference boxes for the 32 new-instance queries.

- For active-instance queries, we use the latest predicted bounding box of the corresponding instance (transformed into the current frame's coordinate system) as its reference box.

The decoder consists of 5 layers, where self-attention among queries enables interaction between new and active queries, allowing them to specialize in discovering unseen objects and refining known ones, respectively.

**Prediction Heads.** We use two separate but structurally similar prediction heads for the outputs corresponding to new-instance and active-instance queries. Each head

ground truth   global view input   egocentric view input

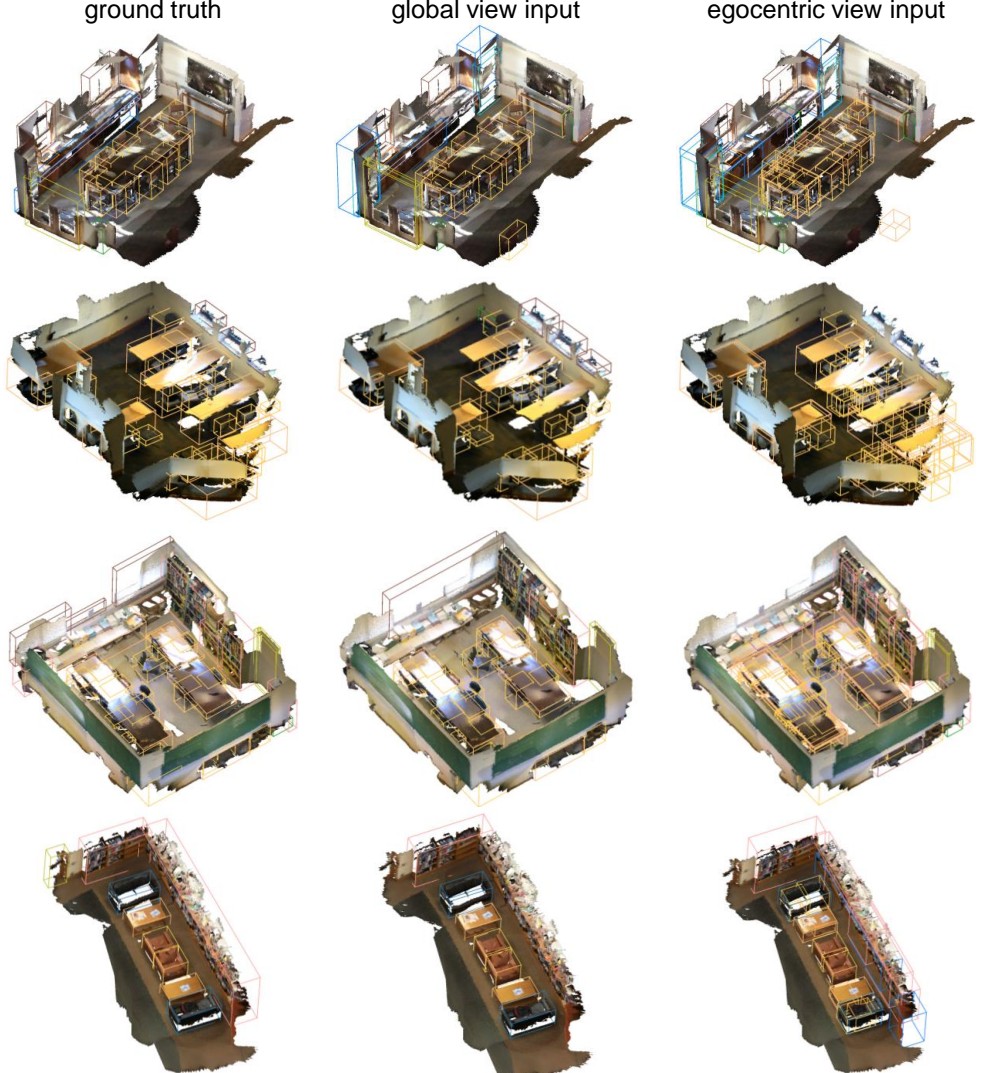

*Figure 9.* Quantitative comparison under different views of input. The left is the ground truth, the middle is the detection of the SOTA method V-DETR when inputting from a global view (ScanNet v2 benchmark), and the right is that when inputting from an egocentric view (obtained by aggregating frame-by-frame predictions using the method mentioned in Section A.2)

contains independent MLP branches for predicting category, center offset, and size offset. Each MLP follows a `Linear-LayerNorm-ReLU-Linear` structure.

- For an output $o_i^{ni}$ from a new-instance query, the category head predicts an 18-dimensional logit vector (for the 18 categories). The center and size heads predict residuals relative to the query's reference box.

- For an output $o_k^{ai}$ from an active-instance query, the category head predicts a single logit score indicating the presence or absence of the instance in the current frame. The center and size heads also predict residuals relative to the instance's previous bounding box (the reference box).

This asymmetric design reflects the distinct roles of the two query types in the temporal detection process.

### B.2. Memory Bank Management

The memory bank serves as the core repository for aggregating information across time, storing both intertemporal scene features and instance states. Its design prioritizes efficiency and scalability for long sequences.

**Feature Point.** The memory bank accumulates the 3D feature points extracted from past frames after transforming them into a unified global coordinate system. To prevent unbounded memory growth as the sequence length increases, we apply voxel downsampling with a voxel size of $8\,\mathrm{cm}$ to

the entire feature point set whenever new points are added. This merges spatially redundant features, significantly reducing storage overhead. In typical indoor environments, this strategy ensures that the memory bank's growth slows and eventually plateaus as the explored space becomes saturated. To empirically validate the efficiency of our memory bank design, we monitor its storage overhead on the **longest** sequence from our test set. Figure 10 illustrates the growth in the memory overhead as the frame index increases. The blue curve corresponds to the actual memory footprint of our method. For reference, the black dashed line depicts the hypothetical growth if all features were stored without downsampling. Our controlled storage strategy yields a memory footprint that remains manageable even for long sequences. At its peak, the memory bank occupies approximately 20 MB. More importantly, the growth rate slows progressively as the explored scene becomes saturated, demonstrating that the overhead scales sub-linearly with sequence length and is bounded in practice. It is a critical property for long-term embodied operation.

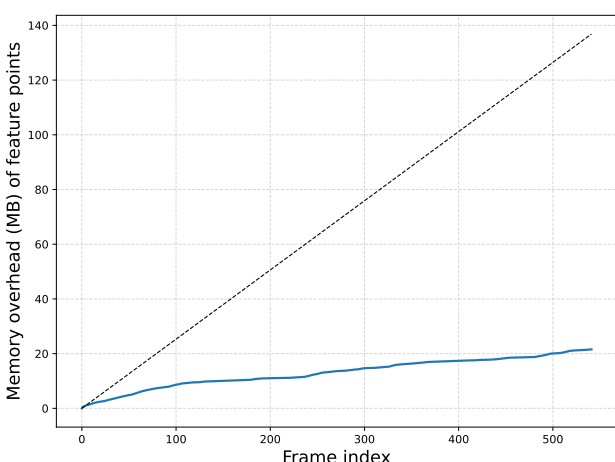

*Figure 10.* Memory overhead of feature points

**Instance.** For each instance, the memory bank stores its latest feature representation, its bounding box in the global coordinate system, and a lifetime counter initialized to 5. The lifetime mechanism governs instance persistence:

- If an active-instance query fails to detect its corresponding object in the current frame (*i.e.,*, its predicted presence confidence is below the threshold), the instance's lifetime is decremented by 1.

- If the instance is successfully detected, its lifetime is reset to the initial value 5.

- An instance is removed from the memory bank once its lifetime reaches 0.

The number of instances stabilizes over time in our memory bank, as shown in Figure 11.

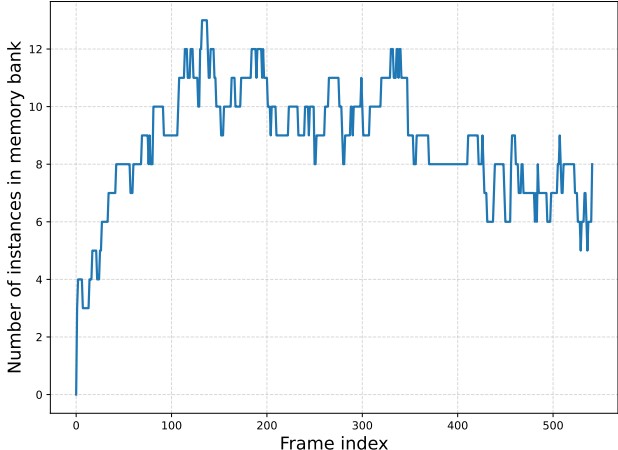

*Figure 11.* Number of instances in the memory bank

**Efficient Retrieval via Frustum Culling.** When querying the memory bank (for either feature points or instance states in Section 4.1 & Section 4.2), we only retrieve items within the current FoV. This is efficiently implemented by projecting the 3D coordinates of stored points/boxes into the current camera's frustum coordinates using the camera intrinsics. This spatial pruning limits the computational cost of feature aggregation and query construction to scale with the visible scene complexity rather than the total explored area. Collectively, these strategies ensure that *Embodied-DETR* operates efficiently per frame and remains scalable for long exploration sequences.

### B.3. Training and Inference

Our implementation is based on the MMDetection3D framework (Contributors, 2020). We utilize the MinkowskiEngine (Choy et al., 2019) as the backend for efficient 3D sparse convolutions, and rely on MMCV (Contributors, 2018) and PyTorch3D (Ravi et al., 2020) for other 3D operations and utilities.

**Data Preprocessing.** Following common practices in indoor 3D detection, we apply the following preprocessing steps to each input frame:

- Transform the point cloud using the axis-aligned pose rotation matrix provided by ScanNet.

- Filter points that are more than $5.12 \, \mathrm{m}$ from the camera center along any axis.

- For RGB values, we normalize them by subtracting the mean $(109.8, 97.2, 83.8)$ and dividing by 255.

*Table 6.* Comparison of *Embodied-DETR* with and without NMS on the *Embodied-Det* test set. The best results are shown in **bold**.

| Method | NMS | Frame-Level | | Scene-Level | | Consistency | | Stability | |
|---|---|---|---|---|---|---|---|---|---|
| | | mAP$_{25}$↑ | mAP$_{50}$↑ | mAP$_{25}$↑ | mAP$_{50}$↑ | mALD↓ | mASD↓ | mCDR↑ | mCMR↓ |
| *Embodied-DETR* | | 59.28 | **44.00** | 67.99 | 50.92 | 0.0399 | 0.7158 | **58.03** | **34.52** |
| *Embodied-DETR* | ✓ | **59.29** | 42.90 | **69.57** | **52.28** | **0.0271** | **0.6895** | 54.23 | 39.37 |

- The ground-truth bounding box for a visible object is defined as the tightest axis-aligned box enclosing its corresponding mesh vertices.

**Data Augmentation.** During training, we apply the following common augmentations:

- **Random flip**: Flip the frame along the x-axis and y-axis, each with a probability of $50\%$.

- **Random rotation**: Rotate the frame around the z-axis by a random angle uniformly sampled from $[-5°, 5°]$.

- **Random translation**: Translate the frame by a random offset sampled from $[-0.1, 0.1]$ meters along each axis.

- **Random scale**: Scale the frame by a random factor uniformly sampled from $[0.9, 1.1]$.

**Training Hyperparameters** We train *Embodied-DETR* using the AdamW optimizer (Reddi et al., 2019) with an initial learning rate of $3 \times 10^{-3}$, weight decay of $0.01$, a linear warmup, and a cosine annealing schedule thereafter. We train for 20 epochs, with a batch size of 16 across all experiments.

For the series of training strategies mentioned in Section 4.4, the specific settings are as follows:

- **Incremental Sequence Learning**: The clip length per sample is progressively increased: 2 frames for epochs 1–4, 4 for epochs 5–8, 8 for epochs 9–12, and 16 for epochs 13–20. When the sampled clip length is smaller than the batch size, multiple random clips are packed into a single mini-batch.

- **Spatial Data Sampling**: We pre-compute the view overlap ratio between all frames. At the beginning of each epoch, we apply FPS with a random seed to select a set of anchor frames from each sequence. For each anchor frame, we randomly sample clip_length $- 1$ additional frames from those having an overlap ratio greater than 0.1 with it to form a training clip.

- **Group Query**: The number of group $G$ is set to 8.

**Matching and Loss Function.** For matching, we first assign ground-truth objects to their corresponding active-instance queries using the stored instance ID. The remaining unmatched ground-truth objects are then assigned to the new-instance queries $\mathcal{Q}^{\mathrm{ni}}$ using the standard Hungarian matching algorithm as in DETR, following the cost matrix formulation of V-DETR. The loss functions $\mathcal{L}_{\mathrm{reg}}$, $\mathcal{L}_{\mathrm{iou}}$ are kept identical to those used in V-DETR.

**Post-processing.** We apply standard post-processing to the per-frame predictions during both training (for memory bank updates) and inference: confidence thresholding at 0.01, followed by Non-Maximum Suppression (NMS) with an IoU threshold of 0.25. During training, an additional IoU threshold of 0.25 is used to filter out low-quality predictions before adding them to the memory bank. Notably, as a property of the DETR architecture, our method yields coherent results even **without NMS** in the temporal setting, as shown in Table 6. Applying NMS primarily enhances the spatial consistency of predictions, significantly reducing location and size deviation (mALD and mASD) and improving holistic scene understanding (higher scene-level mAP). However, it also leads to a slight drop in the most stringent frame-level accuracy (mAP$_{50}$) and temporal stability (mLCDR/mLCMR). This indicates that while NMS helps prune spatially redundant boxes, it can also suppress potential objects from different viewpoints in sequential data. The overall performance balance suggests that our end-to-end framework, even without NMS, already achieves competitive accuracy while maintaining superior temporal coherence, highlighting its inherent capability to handle duplicate suppression through learned instance memory.

## C. Experiment Setting Details

**Hardware.** All experiments are conducted on a server equipped with an Intel(R) Xeon(R) Gold 5218R CPU @ 2.10GHz, 512GB RAM, 8TB SSD, and 4 NVIDIA GeForce RTX 3090 GPUs.

**Attempts to Adapt V-DETR for Temporal Detection.** We also attempted to adapt the SOTA single-frame detector, V-DETR (Shen et al., 2024), as a temporal baseline for *Embodied-Det* using heuristic strategies. Two adaptations were explored: (1) *Point Cloud Stacking*: aligning and concatenating point clouds from a sliding window of past

frames as input; (2) *Post Fusion*: applying Kalman filtering and sliding-window fusion to the per-frame outputs. Neither approach yielded consistent improvements. The best variant only slightly reduced the average location and size deviation (mALD: 0.0535 vs. 0.0675; mASD: 0.7704 vs. 0.8266) while degrading other metrics. Given the marginal and unstable gains, we concluded that rule-based adaptations are insufficient to equip a single-frame detector with robust temporal perception, and such engineering falls outside the core focus of this work. Therefore, these results are not included in the main comparison.

**Implementation Details of Comparison Methods.** We evaluate several classical and SOTA 3D object detection methods on our *Embodied-Det* benchmark, as listed in Table 1 of the main text. To ensure a fair and unified comparison, we base our implementations on the same MMDetection3D framework. For methods already available in MMDetection3D (*e.g.*, VoteNet, H3DNet, GroupFree3D, FCAF3D, etc.), we use the official codebase and configuration files. For others, we carefully re-implement them within the same framework. We verify that the re-implemented models achieve performance on the original ScanNet v2 benchmark (global mesh input) that is comparable to, or matches, the results reported in their respective original publications.

All models are trained on our *Embodied-Det* training set. The performance of each method is evaluated using the final model checkpoint from the last epoch on our test set. While we largely preserve the hyperparameters and architectural choices from their original ScanNet configurations, the following key adaptations are made for our benchmark setting:

- **Training Epochs**: We fix the training schedule to 20 epochs for all methods, which is the same as ours. Although this is fewer than the typical 100+ epochs used for training on the complete ScanNet mesh data, the total number of training *iterations* is actually higher due to our significantly larger number of training frames (over 180k vs. ScanNet's 1.2k training scans). This extended iteration count is necessary given the increased difficulty of our sequential, partial-observation task.

- **Pre-processing for UniDet3D**: UniDet3D relies on pre-computed geometric superpoints for its feature grouping and proposal generation. Following its original pipeline, we pre-generate superpoints for every frame in our dataset. As noted in the main paper, the per-frame RGB-D point clouds are noisier and less complete than the smoothed, reconstructed meshes used in the original ScanNet benchmark. This unavoidably degrades the quality of the superpoint segmentation. We deliberately retain this effect as it reflects a

realistic sensor input condition. All baselines receive the same raw point clouds as input, ensuring a fair comparison under equally challenging sensor data.

