# OpenReview forum: "Embodied-DETR: End-to-End Temporal 3D Object Detection in Egocentric Views"
_ICML.cc/2026/Conference — ICML 2026 regular_

### Official Review · Reviewer_7CYg · 2026-03-07

**Soundness:** 4
**Presentation:** 3
**Significance:** 4
**Originality:** 3
**Overall Recommendation:** 5
**Confidence:** 5

**Summary:**

This paper investigates the problem of temporal 3D object detection based on first-person perspective sequences in embodied perception scenarios. Unlike traditional 3D detection methods that rely on complete scene reconstruction, embodied agents typically only obtain sequential but partial observations. Therefore, the authors propose a novel model, Embodied-Det, which reconstructs the ScanNet dataset into first-person temporal sequences and introduces temporal metrics to evaluate detection stability and continuity. Methodologically, the paper introduces the Transformer-based end-to-end detection framework Embodied-DETR, comprising two core modules: Scene-Aware Feature Aggregation (SFA) for scene-level temporal information aggregation using historical frame features; and Instance-Aware Query Embedding (IQE) for maintaining instance-level queries, thereby distinguishing newly appearing objects from existing ones to preserve detection temporal consistency. Experimental results demonstrate that Embodied-DET outperforms multiple existing indoor 3D detection methods in both detection performance and temporal stability metrics.

**Compliance With Llm Reviewing Policy:**

Affirmed.

**Key Questions For Authors:**

1. Regarding Comparisons with 2D Sequential Detection Work
The memory bank + dual-query mechanism you propose in your paper shares core conceptual similarities with classic works in video object detection/multi-object tracking. While your application focuses on 3D point clouds and embodied intelligence, could you more clearly discuss these connections and distinctions within the paper? For instance, beyond the shift in input modality, what unique challenges did you encounter in the problem's essence (e.g., data sparsity, viewpoint variations, occlusion handling)? How does your model design innovate to address these challenges?
2. Implementation Details of the Adaptive Threshold Mechanism
The adaptive threshold section is described as: “During training, we record the confidence distribution of all positive predictions. At the end of each epoch, we select the threshold that maximizes the detection F1 score on the training set and use it for memory bank management in the next epoch.” How is the F1 score on the training set computed? Is it based on frame-by-frame matching results or instance-level tracking accuracy? Additionally, does this epoch-spanning global statistical update interact with model training within the same epoch, potentially affecting training stability? For example, could using one epoch's distribution to determine the next epoch's threshold introduce oscillation?
3. Discussion on Normalized Error Metrics (mALD/mASD)
In Equations (2) and (3), positional errors and dimensional errors are normalized. This normalization method may amplify minute absolute errors into substantial normalized errors for small objects or objects extremely distant from the camera. Does the dataset contain such extreme cases? Have other more robust normalization methods been considered?

**Limitations:**

Yes

**Strengths And Weaknesses:**

The technical approach is robust, with a rigorously constructed benchmark, well-designed evaluation metrics, and comprehensive experimental design. It compares multiple state-of-the-art methods, and ablation studies thoroughly validate the effectiveness of each component. The appendix provides rich details and strong reproducibility. Limitations of the method are also discussed. The overall structure is clear, the logic flows smoothly, and the core ideas are easy to grasp. The construction of the Benchmark and the description of model details are well-defined. Embodied-DETR provides a standardized approach for this research direction. The memory-enhanced DETR paradigm proposed by Embodied-DETR offers an effective reference for online perception and provides significant inspiration for subsequent related studies. Although focused on indoor 3D detection, the methodological framework can be extended to broader applications. Innovation is primarily embodied in the combination of online learning, first-person perspective, partial observation, and the DETR+Memory Network+Dual Query mechanism. The dual query design is exceptionally clear, achieving an excellent balance between exploration and tracking. While the core design philosophy shares similarities with 2D video object detection, its adaptation to 3D point clouds and embodied scenarios constitutes sufficient innovation.

---

> ### Author Rebuttal · Authors · 2026-03-31
>
> We sincerely thank you for the positive assessment and insightful questions. We address the questions below.
>
> ## **Q1: Connection to 2D Sequential Detection**
>
> The main connection between prior 2D video detection frameworks and us is the core contribution of DETR, which treats query embeddings as the starting point for all detections. However, ​3D embodied perception introduces substantial differences​, which fundamentally affect both the problem formulation and model design.
>
> **Key challenges** in 3D embodied settings include a significantly larger search space, which leads to more difficult optimization and higher computational complexity. In practice, without our proposed training strategies, the temporal module struggles to learn meaningful patterns from noisy single-frame predictions. Besides, severe noise in annotations and sensor data requires additional processing, such as point cloud registration and filtering for data construction.
>
> At the same time, 3D also provides **unique opportunities**. It shows stronger spatial locality, which we leverage for robust and scalable memory management without relying on fixed sequence lengths. We can construct a flexible sequence for training where spatial relationships enable richer pseudo-sequences beyond strictly sequential viewpoints. Physically meaningful 3D bounding boxes are invariant to perspective distortions (e.g., scale changes in 2D images).
>
> We also conducted a series of additional design **explorations** aimed at further improving performance, including:
>
> * Position-guided query initialization based on pose-aligned historical instances,
> * Enhance active-instance queries by direct fusion/recursive fusion/attention-based fusion of aggregating instance features within multi-frame historical information.
> * Decoupling the first decoder layer to separately process different types of queries.
>
> Despite being well-motivated, these designs did not lead to experimental improvements. We believe this further highlights the unique challenges of embodied 3D perception and suggests that effective temporal modeling in this context is non-trivial.
>
> ## **Q2: Adaptive Threshold Mechanism**
>
> The F1 score is computed at the ​**instance level**​, based on detection and association results following the same pipeline as inference.
>
> Specifically, for each frame during training, predictions are converted into instances and matched to ground truth to determine positive/negative labels. At the end of each epoch, we aggregate all instances and evaluate **100 candidate thresholds** (uniformly sampled from [0, 1) with step size 0.01). We calculate the F1 values for each candidate, and the threshold that maximizes the F1 score is selected and applied for memory bank management in the next epoch. To ensure stability, the threshold is ​**kept fixed within each epoch**​, and **updated only once per epoch** based on aggregated statistics.
>
> As the reviewer noted, using the previous epoch’s distribution may introduce a slight lag. However, alternative strategies (e.g., per-step updates) would significantly reduce sample size and introduce high variance. Empirically, we observe that the threshold **always converges to a stable value** and often remains unchanged in later training stages, indicating that this mechanism is sufficiently robust in practice.
>
> ## **Q3: Normalized Error Metrics (mALD / mASD)**
>
> We adopt normalization following a principle similar to IoU-based evaluation: ​**relative errors are more meaningful than absolute errors**​.
>
> While normalization can amplify small absolute errors for small or distant objects, we consider this behavior ​**intentional and practically meaningful**​. In embodied scenarios, small objects often require ​**high localization precision**​, and even minor deviations can lead to task failure. This is also consistent with standard detection metrics, where small localization errors can significantly affect IoU and AP.
>
> Regarding the dataset, our Embodied-Det does not exhibit extreme pathological cases where normalization becomes unstable. In practice, the proposed metrics behave consistently and provide a meaningful measure of ​temporal stability and geometric accuracy​. We believe these normalized metrics faithfully capture the challenges of ​**precise and stable perception**​, particularly for small objects, which remain an important direction for future research.

---

> > ### Author Rebuttal · Reviewer_7CYg · 2026-04-03
> >
> > The authors have thoroughly addressed all three of my questions:
> > Q1: They clearly articulated the unique challenges facing 3D embodied perception and how their design capitalizes on opportunities specific to 3D. Their candid discussion of failed design attempts further enhances the paper’s persuasiveness.
> > Q2: They provided detailed implementation specifics (instance-level F1 calculations, 100-candidate search, updates per epoch) and addressed stability concerns by demonstrating empirical convergence to stable values.
> > Q3: They theoretically argued that relative error is more meaningful than absolute error in embodied scenarios and noted that small objects require high precision.

---

> > > ### Author Response · Authors · 2026-04-03
> > >
> > > We sincerely thank you for the positive acknowledgement and valuable feedback. We will incorporate the clarifications regarding these points into the final manuscript to further improve its clarity and completeness.

---

### Official Review · Reviewer_nQM7 · 2026-03-10

**Soundness:** 2
**Presentation:** 3
**Significance:** 2
**Originality:** 2
**Overall Recommendation:** 4
**Confidence:** 3

**Summary:**

This paper addresses the task of online egocentric 3D object detection for embodied agents. The work proposes a new evaluation protocol, Embodied-Det, which reorganizes ScanNet v2 into sequential frames with temporal metrics. It further introduces Embodied-DETR, a framework that utilizes scene-aware feature aggregation and instance-aware query propagation to maintain detection continuity across frames. While the temporal metrics show improvement over static baselines, there are concerns regarding the technical rigor of the evaluation.

**Compliance With Llm Reviewing Policy:**

Affirmed.

**Final Justification:**

With the authors' detailed response and supplementary experiments, the manuscript is now much more convincing. Since my major concerns have been resolved and no obvious weaknesses remain, I am happy to upgrade my rating to Weak Accept.

**Key Questions For Authors:**

Please refer to the weakness I mentioned.

**Limitations:**

The authors have partially addressed the limitations, but the discussion remains somewhat superficial.

Suggestions for Improvement:
Technical Limitations:
- While the authors correctly identify the "static environment" constraint, the discussion lacks a deep dive into the failure modes of the temporal memory mechanism. Specifically, how does the model handle long-term occlusions or rapid, jerky camera movements that might cause feature alignment errors? A discussion on the scalability limit of the memory bank for truly long-horizon exploration would also be beneficial.

Dataset Bias & Generalization:
- The benchmark is built entirely on ScanNet v2, which uses specific RGB-D sensors and exhibits specific indoor biases. The authors should discuss how the noisy nature of real-world point clouds might affect the model's robustness when deployed on different hardware.

**Strengths And Weaknesses:**

Strengths:
- Pioneering Benchmarking: Reorganizing ScanNet v2 into a sequential online task fills a gap between static 3D vision and robotic perception. The proposed metrics (mCDR, mCMR) are well-suited for evaluating tracking-like stability in detection.
- Effective Query Design: The separation of "new-instance" and "active-instance" queries is a logical approach to handle the appearance and persistence of objects in an online stream.
- Efficiency: The model achieves competitive latency (112ms), making it potentially viable for real-time embodied applications.

Weaknesses:
- Scene-level Metric Logic: In Section 3.2, the confidence of a fused cluster box is defined as the sum of constituent boxes' scores. This design is logically flawed as it rewards models that produce high-density redundant detections, potentially inflating the mAP without improving precision. A mean or max confidence may be more robust?
- Confounding Factors in Temporal Analysis: The "temporal accumulation" trend in Figure 4 is not definitively proven. In ScanNet trajectories, a higher frame index often correlates with higher cumulative scene coverage. Without a control experiment, it is unclear whether the gain stems from the model's memory or simply from the increased observability of the environment.
- Baseline Competitiveness: Although the authors explored rule-based temporal adaptations in the appendix, the main paper lacks comparison with existing learnable temporal 3D detectors. Comparing primarily against static detectors makes the proposed method's contribution difficult to isolate.
- Experimental Rigor: Training all ablation variants on only 1/4 of the training set is a significant limitation. I It is unclear whether the observed gains are robust or if they are artifacts of the reduced data regime.

---

> ### Author Rebuttal · Authors · 2026-03-31
>
> We sincerely thank you for the constructive feedback and we address the concerns below.
>
> ## **W1: Scene-level Metric Logic**
>
> We emphasize that the **sum of confidences** is intentionally designed to reflect *temporal consistency* rather than single-frame optimality.
>
> Using mean or max confidence can lead to undesirable behavior. For example, if a model produces consistent correct detections (e.g., confidence 0.9) across many frames but makes a single high-confidence error (e.g., 0.95), mean/max would retain the erroneous prediction and penalize scene-level mAP disproportionately. In contrast, summation better reflects ​**prediction frequency**​, distinguishing between occasional vs. persistent errors.
>
> Moreover, all methods apply ​**NMS**​ for each frame prediction, which mitigates redundant detections. Therefore, the concern about rewarding dense duplicate predictions is largely alleviated.
>
> ## **W2: Confounding Factors in Temporal Analysis (Fig. 4)**
>
> We clarify that each point in Fig. 4 is computed ​**independently per frame index**​. Specifically, for index $x$, mAP is computed over all sequences’ $x$-th frames only, without accumulating previous frames.
>
> To control for dataset bias, we include a **​temporal-free baseline(SFD)​**, whose performance shows no correlation with frame index. It means that the data with the higher frame index does not carry more valuable information. In contrast, temporal variants (SFD+SFA, SFD+IQE, and SFD+SFA+IQE) exhibit consistent positive correlation (Pearson r>0, p<0.05, Figure 4 only retains the baseline and the full model for clearer visual effects), indicating that gains stem from **temporal modeling of scene and instance information rather than increased observability**.
>
> ## **W3: Baseline Competitiveness**
>
> To the best of our knowledge, there are **no prior temporal 3D detectors** designed for ​*online egocentric indoor settings*​, as we mentioned at the end of the Related Work section. Existing indoor detectors assume global scene access. Therefore, comparisons with single-frame detectors are currently the only feasible and fair setting. Our work aims to establish this new problem and benchmark. We hope our benchmark will facilitate future comparisons in this direction.
>
> ## **W4: Training Data in Ablations**
>
> Using a subset of large-scale datasets for ablations is a **widely adopted practice** in 3D detection (e.g., UVTR, MSMDFusion, VoxelNeXt, MAESTRO). Given that Embodied-Det contains \~180k training frames, we follow this standard protocol. Importantly, all methods are evaluated on the ​**full validation set**​, and we observe consistent trends across settings, suggesting robustness of conclusions.
>
> ## **L1: Analysis of the Failure of Temporal Mechanism**
>
> We further clarify that our Memory Bank Management (Appendix B.2) is designed to handle both memory failure and long-horizon scenarios.
>
> **(a) Robustness to memory failure (e.g., occlusion, rapid camera motion):**
> When retrieving historical information, we **filter out invisible content** (e.g., fully occluded objects or out-of-view points). Therefore, if temporal memory becomes unreliable, the model naturally **degrades to a single-frame detector** rather than introducing negative interference.
> We validate this by disabling **all** memory at inference:
>
> |variant|mAP25|mAP50|
> |-|:-:|:-:|
> |SFD|45.67|28.48|
> |SFD+SFA+IQE w/o memory|45.97|29.82|
>
> The comparable performance shows that the model can ​**robustly handle complete memory failure**​.
>
> **(b) Long-horizon scalability:**
> We control memory growth via:
>
> * **point density regulation** (bounded in enclosed indoor environments),
> * **instance lifecycle management** (removing disappeared/false objects), and
> * **visibility-based filtering** during retrieval.
>
> We further simulate long-horizon sequences (up to 10k frames) by trajectory replay, and record the performance as the frame index increases.
>
> |Frame|100|1k|3k|5k|7k|10k|
> |-|:-:|:-:|:-:|:-:|:-:|:-:|
> |**mAP50**|58.04|58.17|58.11|58.90|59.00|59.01|
> |**$N_{points}$**|13k|36k|36k|36k|37k|37k|
> |**$N_{ins}$**|9|12|7|7|7|8|
>
> $N_{points}$ and $N_{ins}$ represent the quantities of feature points and instances in the memory bank, respectively. Results show our stable accuracy and bounded memory size over time.
>
> Overall, our design ensures robustness to temporal failures and scalability to long-horizon inference. We agree that further analysis is valuable and will include it in our manuscript.
>
> ## **L2: Dataset Bias & Generalization**
> We acknowledge that ScanNet v2 reflects ​indoor RGB-D biases​. However:
>
> * Indoor scenes and RGB-D sensors align with **common** embodied robotics settings.
> * ScanNet already contains **real-world noise** (sensor noise, systematic error, less accurate measurement), which challenges the robustness of models.
>
> Meanwhile, we provide comprehensive data construction and performance evaluation benchmarks that can be applied to specific hardware or scenarios to adapt to specific vertical domains.

---

> > ### Author Rebuttal · Reviewer_nQM7 · 2026-04-04
> >
> > The authors have addressed most of my concerns from a theoretical perspective. However, some weaknesses (e.g., Weakness 1 and 2) lack detailed experimental validation, which limits the persuasiveness of the response. Regarding the explanation for Limitation 1: Analysis of the Failure of Temporal Mechanism, the authors only explored the scenario of disabling all memory. In this case, the proposed method shows no significant advantage over the baseline. The response still lacks a deep dive into handling long-term occlusions or rapid, jerky camera movements that might cause feature alignment errors. Therefore, I am inclined to maintain my current score and will refer to the other reviewers' opinions.

---

> > > ### Author Response · Authors · 2026-04-06
> > >
> > > We thank the reviewer for the follow-up and provide additional experimental evidence and clarifications below.
> > >
> > > ## **W1: Scene-level Metric Logic**
> > >
> > > We conduct **quantitative and qualitative** comparisons of three confidence aggregation strategies, **sum, max, and mean**, across all baselines. Quantitative results (mAP25/mAP50) are shown below:
> > >
> > > |Operation|VoteNet|H3DNet|GroupFree3D|FCAF3D|TR3D|V-DETR|UniDet3D|Ours|
> > > |:-:|:-:|:-:|:-:|:-:|:-:|:-:|:-:|:-:|
> > > |mean|40.88/25.69|43.36/29.27|43.98/28.40|51.39/38.93|53.77/38.57|56.04/40.98|30.31/15.21|59.52/44.26|
> > > |max|51.33/35.55|55.38/40.73|50.77/32.91|57.04/42.36|60.21/43.95|60.42/44.61|43.26/29.63|66.10/50.06|
> > > |sum|53.88/36.59|58.26/40.51|52.39/34.45|58.98/44.68|61.46/45.78|66.88/50.28|55.07/35.91|69.57/52.28|
> > >
> > > **Sum almost always achieves the best performance across all methods**, and ours remains the top-performing approach under all strategies.
> > >
> > > Qualitatively (see [anonymous material](https://anonymous.4open.science/r/AnonymousFiles-B627/nQM7/Figure_1.png)), **sum suppresses low-frequency false positives**, while ​**max/mean are more affected by occasional errors**​.
> > >
> > > Thus, **sum better reflects consistent scene-level understanding** and is both experimentally and conceptually justified.
> > >
> > > ## **W2: Confounding Factors in Temporal Analysis (Fig. 4)**
> > >
> > > We provide **extended results in the [anonymous material](https://anonymous.4open.science/r/AnonymousFiles-B627/nQM7/Figure_2.png)**, extending Figure 4 in our manuscript to include all four variants: **SFD, SFD+SFA, SFD+IQE, and SFD+SFA+IQE**.
> > >
> > > We further clarify the experimental protocol. For predictions over 300 validation sequences, we construct evaluation subsets **independently for each frame index**. Specifically, for a given index *x*, we compute mAP over a subset consisting of the *x*-th frame from each sequence (i.e., 300 frames in total). For example:
> > >
> > > * At *x = 1*, the reported mAP is computed over all first frames;
> > > * At *x = 8*, it is computed over all 8th frames and frames with index < 8 are not included.
> > >
> > > Importantly, each input at frame *x* contains **only the current frame’s point cloud**, without extra accumulated observations. The storage and utilization of historical information depend entirely on the model itself.
> > >
> > > Under this protocol, the **temporal-free baseline (SFD)** shows **no correlation between performance and frame index**, indicating that the data itself does not become inherently more informative at higher indices. In contrast, all temporal variants (**SFD+SFA, SFD+IQE, SFD+SFA+IQE**) exhibit a **positive correlation with frame index**, demonstrating that the performance gains arise from leveraging **temporal information**(i.e., historical **scene-level and/or instance-level context**).
> > >
> > > ## **L1: Analysis of the Failure of Temporal Mechanism**
> > >
> > > We clarify several potential misunderstandings.
> > >
> > > First, regarding the *complete memory failure* experiment (**SFD vs. SFD+SFA+IQE w/o memory**), the reviewer noted that *“the proposed method shows no significant advantage over the baseline.”*
> > > We emphasize that this result is **expected and consistent** with our design.
> > >
> > > Our method is designed to leverage historical information. When all historical information is removed, it becomes functionally equivalent to the single-frame detector (SFD). Therefore, no temporal method can exhibit a significant advantage over a single-frame baseline under this condition.
> > >
> > > The purpose of this experiment is to demonstrate **robustness under memory failure**: when historical information is unavailable, the model **gracefully degrades** to a single-frame detector without introducing additional negative effects.
> > >
> > > ---
> > >
> > > We further analyze **feature alignment errors** based on their causes:
> > >
> > > * **Nominal conditions (no pose error):**
> > >   Temporal alignment is performed using camera poses, transforming all frames into a unified coordinate system. Thus, *occlusions* or *rapid camera motion* do not affect alignment or introduce errors, and do not lead to measurable performance changes.
> > > * **Moderate degradation (pose estimation errors):**
> > >   Pose inaccuracies may introduce misalignment in the memory bank. This is a valid concern, and we refer to our response to **Reviewer Qmvc W1**, where we provide experimental analysis of the impact of **pose noise and real-world estimation errors**.
> > > * **Severe failure (pose breakdown):**
> > >   If pose estimation becomes unreliable, the model **disables memory interaction** and ignores historical information, reducing to a single-frame detector without additional degradation. This behavior is validated by the rebuttal experiment.
> > >
> > > ---
> > >
> > > Finally, as in most temporal 3D perception tasks (e.g., 3D MOT, 4D segmentation, temporal SSC), we assume **reliable poses as input** and focus on their utilization. Extreme cases where poses are unavailable are beyond scope. Nevertheless, we analyze such misalignment and show our method remains ​**robust under such conditions**​.

---

### Official Review · Reviewer_yyaW · 2026-03-12

**Soundness:** 3
**Presentation:** 3
**Significance:** 3
**Originality:** 2
**Overall Recommendation:** 3
**Confidence:** 2

**Summary:**

This paper addresses temporal 3D object detection from egocentric views for embodied agents. It introduces Embodied-Det, a benchmark reorganized from ScanNet v2 that provides sequential RGB-D frames with per-frame 3D bounding box annotations and novel temporal evaluation metrics (mALD, mASD, mCDR, mCMR). On this benchmark, the  authors propose Embodied-DETR, an end-to-end transformer framework featuring two temporal modules: SFA, which enhances current-frame features using a spatial memory bank of past observations, and IQE, which constructs queries from previously detected instances to enable multi-frame reasoning. The method achieves state-of-the-art results across detection accuracy and temporal consistency metrics, outperforming seven baseline detectors.

**Compliance With Llm Reviewing Policy:**

Affirmed.

**Key Questions For Authors:**

1. Can MPPNet (Chen et al., 2022) or PTT (Huang et al., 2024) be adapted to indoor egocentric detection and compared on Embodied-Det? Even a best-effort adaptation would clarify the contribution of end-to-end training vs. heuristic temporal fusion.

2.  Can you provide a per-component latency breakdown (backbone, SFA, IQE, decoder, memory bank update) to identify the practical bottleneck?

3. Is the higher mCMR due to competition between active-instance queries and new-instance queries in the cross-attention? Specifically, do active-instance queries dominate attention to features, leaving insufficient signal for new-instance queries in crowded scenes?

**Limitations:**

Yes

**Strengths And Weaknesses:**

1. ALD, ASD, CDR, and CMR are well-defined, theoretically motivated, and address a real gap in prior evaluation protocols that focus only on per-frame detection accuracy. The per-frame mAP correlation analysis provides strong quantitative evidence that the temporal modules enable progressive information accumulation — a compelling result.

2. The appendix provides detailed descriptions of memory bank management (voxel down-sampling, lifetime mechanism, frustum culling), memory growth analysis (Figure 8), and instance count stabilization (Figure 9). These demonstrate genuine engineering rigor and make the system practically viable.

3. The decomposition into scene-level (SFA) and instance-level (IQE) temporal reasoning is intuitive and well-justified: SFA broadens spatial context while IQE maintains identity-specific continuity. The decoupled bipartite matching strategy elegantly handles the asymmetry between discovering new instances and tracking known ones. The ablation study convincingly demonstrates that both modules contribute and their combination is complementary. The per-frame mAP analysis provides compelling evidence that temporal accumulation is occurring.

# Weakness
1. This works links egocentric video and 3D object detection, which is meaningful. However, why do not use the latest egocentric-based benchmarks for dataset construction? The latest egocentric benchmarks (e.g., LookOut: Real-World Humanoid Egocentric Navigation, ICCV 2025) contain richer information like interaction between the first person and people around, the scanNetv2 is more about exocentric video.

2. It states that the efficiency of the proposed Embodied-DETR is one of the advantages, but it does not have the advantage among the compared methods, as shown in Table 1.

3. As far as I can see, the Instance-aware Query Embedding is a memory mechanism that stores features over time, and the SFA module is essentially a standard PointNet++ set abstraction. It does not have strong novelty.

---

> ### Author Rebuttal · Authors · 2026-03-31
>
> We sincerely thank you for the reasonable suggestion and we address the concerns below.
>
> ## **W1: Choice of Dataset**
>
> We agree that recent egocentric benchmarks (e.g., LookOut) are valuable. However, such datasets ​**do not provide instance-level 3D annotations**​, which are essential for 3D object detection. Constructing large-scale, high-quality 3D instance annotations from scratch is too expensive and beyond our resources. Therefore, we build upon ScanNet v2, which provides reliable annotations and has been widely reused in prior temporal 3D perception tasks. We view Embodied-Det as a ​practical first step​, and future work will explore richer egocentric datasets.
>
> ## **W2: Efficiency Claim**
>
> We apologize for the ambiguity. Here, “efficiency” refers to a holistic trade-off between accuracy, temporal modeling capability, and runtime, but not purely latency. Our method achieves ​higher accuracy with near real-time inference. Among DETR-style methods, it is ​comparable in speed to GroupFree3D but significantly more accurate​, and ​both faster and more accurate than V-DETR​. Importantly, another DETR-style method, UniDet3D, relies on ​**precomputed superpoints**​, whose cost (\~300 ms per frame in our benchmark) is not included in the reported runtime but is required for deployment. We will clarify this definition in our manuscript.
>
> ## **W3: Novelty Concerns**
>
> Our primary contribution is to ​**establish embodied 3D detection as a new problem setting**​, including a dedicated benchmark (Embodied-Det), and an end-to-end temporal detection framework.
> Methodologically, while individual components are inspired by prior designs, our key novelty lies in proposing a temporal detection architecture that originates from highly promising DETR and inherits simplicity and end-to-end characteristics, without relying on some complex but vulnerable designs. We emphasize providing new directions and foundations for future research.
>
> ## **Q1: Comparison with Heuristics**
>
> As discussed in Appendix C, we attempted to extend the strong baseline with heuristic temporal fusion, but observed ​limited or negative gains​.
>
> We also made a best-effort attempt to adapt PTT, but did not obtain meaningful improvements. A key reason is the ​**domain gap between outdoor and indoor 3D detection**​:
>
> * Indoor objects exhibit ​**larger intra-class variation**​, making localization more challenging.
> * Lower-quality proposals (due to occlusion/partial views/rich geometric shapes) lead to ​**unreliable ROI features**​, which PTT heavily depends on.
>
> This aligns with the common observation that indoor and outdoor 3D detection methods are often ​**not directly transferable**​.
>
> ## **Q2: Latency Breakdown**
>
> We provide per-component latency (ms):
>
> | Component | Latency
> | - |:-------:|
> | Backbone |   47    |
> | SFA |    9    |
> |IQE|    8    |
> |Transformer Encoder|    3    |
> |Object Prediction|   42    |
> |Others|    3    |
>
> SFA and IQE include memory bank operations, Object Prediction includes Transformer Decoder and the final FFN(as shown in Figure 3 of our manuscript). The main bottlenecks are ​**backbone and Transformer Decoder**​, while temporal modules introduce ​**modest overhead**​.
>
> ## **Q3: Query Competition & mCMR**
>
> We analyze this by the experimental results reported in Appendix B.3. Without NMS, where our method achieves **the best mCMR**, indicating that ​**query competition does not harm new-instance discovery**​.
>
> The slightly worse mCMR in Table 1 is mainly due to ​**NMS over-suppressing valid detections**​, whereas the dense predictor TR3D is less sensitive due to redundancy.

---

> > ### Author Rebuttal · Reviewer_yyaW · 2026-04-07
> >
> > Thank you for the rebuttal. I am still concerned about the novelty. The main contribution is to ​establish embodied 3D detection as a new problem setting​. Could you please elaborate what embodied 3D detection is? what are its main differences from other 3D detection tasks, such as 3D detection in autonomous driving or robotic senarios?

---

> > > ### Author Response · Authors · 2026-04-07
> > >
> > > Thank you for the follow-up question. We further clarify the definition of **embodied 3D detection** and its distinctions from existing 3D detection tasks.
> > >
> > > ---
> > >
> > > ### **What is Embodied 3D Detection?**
> > >
> > > Embodied 3D detection is designed for **embodied agents** operating in real-world environments. The goal is to detect objects from ​**egocentric, first-person observations**​, enabling the agent to understand and interact with its surroundings at the instance level.
> > >
> > > This setting emphasizes two key aspects:
> > >
> > > * ​**Egocentric perception**​: the agent only observes **partial, view-dependent frames** through its onboard sensors.
> > > * ​**Temporal reasoning**​: the agent can leverage **sequential observations** to accumulate spatial and semantic information over time.
> > >
> > > In terms of setup:
> > >
> > > * ​**Data**​: we follow common embodied robotics settings, focusing on ​**indoor environments with RGB-D sensors**​.
> > > * ​**Metrics**​: in addition to standard detection metrics (IoU, mAP), we introduce **temporal consistency and stability metrics** (ALD, ASD, CDR, CMR), which are not captured by frame-wise evaluation.
> > >
> > > ---
> > >
> > > ### **Differences from Existing 3D Detection Tasks**
> > >
> > > #### **(a) Difference from Autonomous Driving 3D Detection**
> > >
> > > The primary difference lies in the ​**indoor vs. outdoor domain gap**​, which leads to fundamentally different problem characteristics:
> > >
> > > * ​**Object distribution**​:
> > >   Indoor objects can appear anywhere with stacking and containment relationships, while outdoor objects are typically distributed on the ground plane, enabling BEV simplifications.
> > > * ​**Sensor modality**​:
> > >   Indoor: RGB-D with dense but partial observations.
> > >   Outdoor: LiDAR with sparse but wide-range coverage.
> > > * ​**Object categories and geometry**​:
> > >   Indoor objects (e.g., furniture) exhibit **high intra-class variation** in shape and size, whereas outdoor objects (e.g., vehicles) are more regular.
> > >
> > > Due to these differences, indoor and outdoor 3D detection methods have largely evolved ​**independently**​. For this reason, our experiments adopt ​**indoor-oriented baselines**​, rather than autonomous driving methods such as PointPillars or CenterPoint.
> > >
> > > #### **(b) Difference from Existing Indoor Robotic 3D Detection**
> > >
> > > Compared to standard indoor 3D detection, our setting differs in both ​**input and output paradigms**​:
> > >
> > > |Difference|Ours|Existing Task|
> > > |-|-|-|
> > > |input|sequential egocentric view RGBD frames|globally reconstructed indoor scenes mesh|
> > > |output|online detection within the current field-of-view|offline detection over the complete scene|
> > >
> > > We provide a schematic diagram in the [anonymous material](https://anonymous.4open.science/r/AnonymousFiles-B627/yyaW/Figure_1.png) (https://anonymous.4open.science/r/AnonymousFiles-B627/yyaW/Figure_1.png) to better illustrate this difference. Existing indoor benchmarks and methods are designed for ​**global, offline scene understanding**​, and do not address the **partial, streaming, and temporally evolving observations** faced by embodied agents.
> > >
> > > This difference introduces additional challenges:
> > >
> > > * **Stronger noise** in RGB-D observations compared to reconstructed meshes
> > > * **Severe occlusion and limited field-of-view**
> > > * **Incomplete geometry** at each timestep
> > >
> > > In our manuscript (Sec. 5.2, ​*Performance Change from Global to Egocentric View*​), we quantitatively demonstrate that these factors significantly degrade the performance of existing indoor detectors, experimentally confirming that ​**embodied 3D detection is a distinct and more challenging setting**​.
> > >
> > > ---
> > >
> > > ### **Why This Setting Matters**
> > >
> > > While temporal modeling has been explored in autonomous driving, ​**there is no established benchmark for temporal 3D detection in embodied indoor scenarios**​. This creates a gap between existing research paradigms and the ​**operational reality of embodied agents**​.
> > >
> > > We position **Embodied-Det** as a ​**first step toward bridging this gap**​, by defining a realistic egocentric, sequential detection setting, and enabling systematic evaluation of temporal perception in embodied environments.
> > >
> > > ---
> > >
> > > We hope this clarifies both the **definition** and the **novelty** of the proposed problem setting.
> > > If you have any further concerns or questions, feel free to let us know.

---

### Official Review · Reviewer_Qmvc · 2026-03-19

**Soundness:** 3
**Presentation:** 3
**Significance:** 3
**Originality:** 3
**Overall Recommendation:** 4
**Confidence:** 3

**Summary:**

This paper addresses the critical gap between traditional 3D object detection, which assumes access to static and globally reconstructed scenes, and the operational reality of embodied agents that must perceive the world through sequential, partial, and often occluded egocentric views. The authors first introduce Embodied-Det, a benchmark that reorganizes the ScanNet v2 dataset into first-person sequences and provides novel metrics to evaluate temporal consistency (measuring prediction jitter) and stability (assessing detection persistence). To tackle these challenges, they propose Embodied-DETR, an end-to-end transformer framework that leverages two core temporal modules: Scene-aware Feature Aggregation (SFA), which pulls context from a global memory bank, and Instance-aware Query Embedding (IQE), which maintains dynamic queries to track and refine object representations over time. Their experiments demonstrate that while existing state-of-the-art detectors suffer significant performance degradation in egocentric settings, Embodied-DETR achieves superior accuracy and stability—beating previous baselines like V-DETR by +3.76 in $mAP_{50}$—while maintaining an efficient processing speed of 112 ms per frame.

**Compliance With Llm Reviewing Policy:**

Affirmed.

**Key Questions For Authors:**

Both the benchmark and the proposed method are currently restricted to static environments. In many real-world embodied scenarios, agents must interact with dynamic objects (e.g., people or pets), and the paper does not yet address how temporal continuity would be maintained when both the observer and the objects are moving independently. How would the author solve this kind of problems in their future works.

**Limitations:**

See weaknesses

**Strengths And Weaknesses:**

Strengths:
- The paper is well written and easy to follow
- The authors introduce Embodied-Det, a large-scale benchmark comprising over 230k frames and 1,513 sequences. By moving beyond standard Average Precision to include Temporal Consistency (mALD/mASD) and Stability (mCDR/mCMR) metrics, they provide a scientifically rigorous way to measure perception quality for real-world robotics where flickering or jittery detections can hinder motion planning.
- The framework introduces Instance-aware Query Embedding (IQE), which propagates dedicated queries for specific physical instances across frames. This enables the model to effectively accumulate information over time

Weaknesses:
- The entire framework for aggregating features and maintaining instance states relies on transforming observations into a "unified global coordinate system" using the "axis-aligned pose" provided by the dataset. In real-world robotics, camera poses are often estimated dynamically and how the memory bank or feature aggregation might degrade if these poses are even slightly inaccurate.

---

> ### Author Rebuttal · Authors · 2026-03-31
>
> We sincerely thank you for the positive evaluation of our work and we address the concerns below.
>
> ## **W1: Dependency on Accurate Camera Poses**
>
> We agree that our framework assumes access to a unified global coordinate system constructed from camera poses. This design **follows** a common practice in multi-frame 3D perception tasks, where pose information is treated as given input (e.g., 4D Segmentation, 3D Multiple Object Tracking, temporal Semantic Scene Completion). Nevertheless, we acknowledge that pose noise is inevitable in real-world robotics and evaluating robustness is important.
>
> To this end, we conducted additional experiments to analyze the sensitivity of our method to pose inaccuracies under two settings:
>
> **(a) Synthetic noise injection**:
> We perturb ground-truth poses with random noise  $Err_\alpha \sim U(-\alpha, \alpha)$.
>
> **(b) Real SLAM-estimated poses**:
> We replace ground-truth poses with dynamically estimated outputs from existing SLAM systems (Co-SLAM(CVPR 2023), CG-SLAM(ECCV 2024), Loopy-SLAM(CVPR 2024)) to simulate practical deployment.
>
> During training, ground-truth poses are used, while during evaluation, noisy poses are applied. All validation data comes from the validation set of SLAM algorithms mentioned above. The detection accuracy under different pose sources is shown in the table.
>
> | Pose Source         | Ground-Truth | $\alpha$=1cm | $\alpha$=3cm | $\alpha$=5cm | Co-SLAM | CG-SLAM | Loopy-SLAM |
> |---------------------|:-:|:-:|:-:|:-:|:-:|:-:|:-:|
> | Frame mAP25 | 86.27 | 85.33 | 85.08 | 83.79 | 85.13 | 86.05 | 85.12 |
> | Frame mAP50 | 75.46 | 75.75 | 75.52 | 73.13 | 74.71 | 74.53 | 76.01 |
>
> The experimental results indicate that performance degradation is marginal under small perturbations.
> Even with ​**real SLAM poses**​, the model maintains competitive performance with only minor drops.
> Larger noise (e.g., 5 cm) does pose a challenge to temporal detection.
>
> Despite not introducing explicit robustness mechanisms, our model demonstrates ​**reasonable tolerance to pose noise**​. A reason is that ​**ScanNet already contains non-negligible pose inaccuracies and real-world noise**​, which implicitly regularizes the model during training.
>
> ## **Q1: Handling Dynamic Environments**
>
> We appreciate this insightful question, and we also explicitly acknowledge this limitation in the paper.
> Our ongoing and planned directions include both **data** and ​**methodology**​:
>
> **(a) Data Perspective**
>
> Current large-scale indoor datasets with 3D annotations lack dynamic objects. To address this, we plan to leverage **simulated environments such as Replica with Habitat-Sim** to generate sequences with controllable dynamic objects, enabling systematic learning and evaluation.
>
> **(b) Method Perspective**
>
> Our framework has inherent advantages for handling dynamic scenes. Instance-aware Query Embedding does **not rely on explicit geometric heuristics** (e.g., cross-frame IoU matching commonly used in tracking). Instead, it performs ​**implicit association via learned representations**​, which is more flexible under complex motion patterns. We are actively exploring extensions to handle dynamic objects, including "**next state prediction mechanisms**​", where instance queries predict future states (location, etc.) based on historical observations. This is designed for enhancing dynamic instance modeling capabilities. It remains fully end-to-end and avoids introducing heuristic motion models (e.g., Kalman filtering).
>
> We believe these efforts are promising for bridging embodied perception and dynamic scene understanding, and we are actively pursuing them.

---

> > ### Author Rebuttal · Reviewer_Qmvc · 2026-04-07
> >
> > Thanks the authors, my issues have been addressed

---

> > > ### Author Response · Authors · 2026-04-07
> > >
> > > Thank you for your kind acknowledgement. We are glad that our additional analyses and clarifications have addressed your concerns. We appreciate your valuable feedback, which has helped us improve the completeness and rigor of the work.

---

### Decision · Program_Chairs · 2026-04-30

**Decision:**

Accept (regular)

**Comment:**

The paper introduces Embodied-Detr, a benchmark for online egocentric 3D object detection with temporal consistency and stability metrics based on ScanNet. Multiple reviewers find the benchmark and model technically solid, well-motivated, and impactful for embodied perception, with three reviewers recommending acceptance (two weak accepts, one accept) and one weak reject primarily due to perceived novelty and dataset choices. The rebuttal and follow-up clarifications substantially strengthen the case for the new problem setting of “embodied 3D detection” and show that temporal modeling yields consistent gains over strong static baselines, while the remaining concerns are either addressed experimentally or are limitations that do not block acceptance. The AC notes that there are still some unsolved concerns, in particular by one reviewer. However, the AC agrees with the generally positive comment and believes that the paper, despite the limitation of not using real egocentric footage, is a step forward for the community and provides a useful tool to study embodied 3D object detection, so the recommendation is to accept.